# WHEN LESS IS MORE: SIMPLIFYING INPUTS AIDS NEURAL NETWORK UNDERSTANDING

## ABSTRACT

How do neural network classifiers behave if you make their inputs simpler? In this work, we propose SimpleBits, a method to synthesize simplified inputs by reducing information content, and measure the effect of such simplification on learning. To measure simplicity, we use the finding that the encoding bit size given by a pretrained generative image model correlates well with the visual complexity of the image. Hence, we minimize the encoding bit size to simplify inputs and investigate the effect of such simplification on neural network behavior in several scenarios: conventional training, dataset condensation and post-hoc explanations. In all settings, we optimize for simplified inputs that still contain the information to learn the classification task. We show that SimpleBits successfully removes superfluous information for tasks with injected distractors and investigate the tradeoff between input simplicity and task performance on real-world datasets. For dataset condensation, we find that inputs can be simplified with only minimal accuracy degradation. Finally, applied post-hoc to normally trained classifiers, SimpleBits provides intuition into reasons for misclassifications.

## 1 INTRODUCTION

A better understanding of the information deep networks learn can lead to new scientific discoveries (Raghu & Schmidt, 2020), inform our understanding of differences between human and model behaviors (Makino et al., 2020) and can serve as powerful auditing tools (Geirhos et al., 2020).

Removing information from the input manually is one way to understand what information content is relevant for learning. For example, researchers have occluded parts of the input or removed specific frequency ranges from the input to see which input regions and frequency ranges are relevant for the network's prediction (Zintgraf et al., 2017; Makino et al., 2020; Banerjee et al., 2021). These ablation techniques use simple heuristics such as removing at random (Hooker et al., 2019) or exploit domain knowledge about interpretable aspects of the input (input regions, frequency range) to create simpler versions of the input and analyze the network's prediction on these simpler inputs.

What if, instead of using heuristics, one would learn how to simplify inputs that contain prediction-relevant information? This way, one could synthesize simpler inputs and gain intuition into model behavior without relying on domain knowledge about what information may be relevant for the network. For this, one needs to define the precise meaning of "simplify an input", including useful metrics for the simplification of the inputs and for the retention of task-relevant information.

In this work, we propose *SimpleBits* – an information-reduction method that learns to synthesize simplified inputs which contain less information but still remain informative for the task. To measure simplicity, we use a finding initially reported as a problem for density-based anomaly detection – generative image models tend to assign higher probability densities and hence lower bits to visually simpler inputs (Kirichenko et al., 2020; Schirrmeister et al., 2020). Here, we use this to our advantage and minimize the encoding bit size given by a generative network trained on a general image distribution to simplify inputs. At the same time, we optimize that the simplified inputs still contain the task-relevant information.

We then investigate what information is retained in the simplified inputs and how the simplification affects network behavior in a variety of settings. We apply *SimpleBits* both in a *per-instance* setting, where each image is processed to be a simplified version of itself, and the size of training set remains

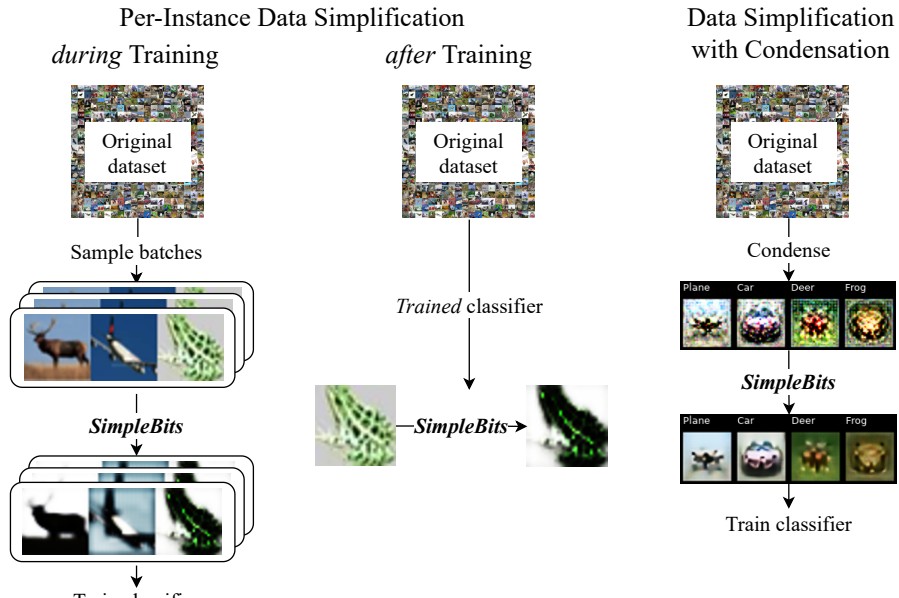

Figure 1: We apply *SimpleBits* to a variety of tasks to aid neural network understanding. As a per-image simplifier, applied during training (**left**), it investigates the trade-off curve between simplification and accuracy. It can also be used as a post-hoc analysis tool after training (**center**), illuminating features of images that are crucial to the trained classifier. When combined with data condensation (**right**), the original data set can be effectively reduced both in size and in complexity.

unchanged, as well as in a *condensation* setting, where the dataset is compressed to only a few samples per class, with the condensed samples simplified at the same time. Applied during training, *SimpleBits* can be used to investigate the trade-off between the information content and the task performance. After training, *SimpleBits* can act as an analysis tool to understand what information a trained model uses for its decision making. Figure 1 summarizes tasks covered in this paper.

Our evaluation provides the following insights in the investigated scenarios:

1. **Per-instance simplification during training.** *SimpleBits* successfully removes superfluous information for tasks with injected distractors. On natural image datasets, *SimpleBits* highlights plausible task-relevant information (shape, color, texture). Increasing simplification leads to accuracy decreases and we report the trade-off between simplifying inputs and task level performance for different datasets.

2. **Dataset simplification with condensation.** We evaluate *SimpleBits* applied to a condensation setting that processes the training data into a much smaller set of synthetic images. *SimpleBits* simplifies these images to drastically reduce the encoding size without substantial task performance decrease. On a chest radiograph dataset (Johnson et al., 2019a;b), *SimpleBits* can uncover known radiologic features for pleural effusion and gender.

3. **Post-training simplification.** For a trained model, *SimpleBits* provides intuition into the prediction-relevant information in an image. For example, by visualizing simplifications of mispredicted images, we can form hypotheses of why these images were mispredicted.

## 2 MEASURING AND REDUCING INSTANCE COMPLEXITY

How to define simplicity? We use the fact that generative image models tend to assign lower encoding bit sizes to visually simpler inputs (Kirichenko et al., 2020; Schirrmeister et al., 2020). Concretely, the complexity of an image $x$ can be quantified as the negative log probability mass given by a pretrained generative model with tractable likelihood, $G$, i.e. $-\log p_G(x)$. This log probabil-

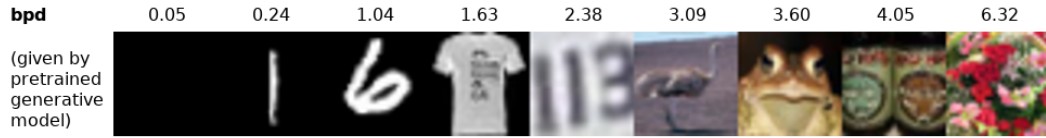

Figure 2: Visualization of the bits-per-dimension (bpd) measure for image complexity, sorted from low to high. Image samples are taken from MNIST, Fashion-MNIST, CIFAR10 and CIFAR100, in addition to a completely black image sample. bpd is calculated from the density produced by a Glow (Kingma & Dhariwal, 2018) model pretrained on 80 Million Tiny Images.

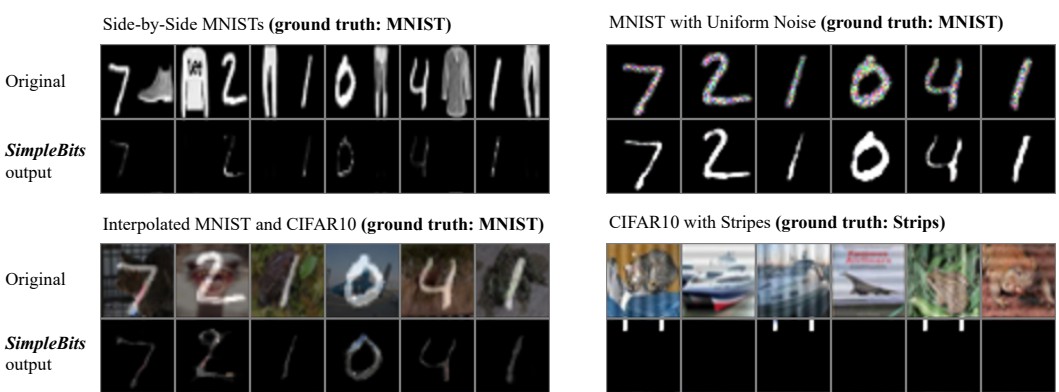

Figure 3: Evaluation of distractor removal on four composite datasets. Shown are the composite original images and the corresponding simplified images produced by *SimpleBits* trained alongside the classifier. *SimpleBits* is able to almost entirely remove task-irrelevant image parts, namely FashionMNIST (**top left**), random noise (**top right**), CIFAR10 (**bottom left** as well as **bottom right**).

ity mass can be interpreted as the image encoding size in bits per dimension (bpd) via Shannon's theorem (Shannon, 1948): $\text{bpd}(\boldsymbol{x}) = -\log_2 p_G(\boldsymbol{x})/d$ where $d$ is the dimension of the flattened $\boldsymbol{x}$.

The simplification loss for an input $\boldsymbol{x}$, given a pre-trained generative model $G$, is as follows:

$$L_{\text{sim}}(\boldsymbol{x}) = -\log p_G(\boldsymbol{x}) \tag{1}$$

Figure 2 visualizes images and their corresponding bits-per-dimension (bpd) values given by a Glow network (Kingma & Dhariwal, 2018) trained on 80 Million Tiny Images (Torralba et al., 2008) (see supp. sec. S1 for other models). This is the generative network used across all our experiments. A visual inspection of Figure 2 suggests that lower bpd corresponds with simpler inputs, as also noted in prior work (Serrà et al., 2020). The goal of our approach, *SimpleBits*, is to minimize bpd of input images while preserving task-relevant information.

We now explore how *SimpleBits* affects network behavior in a variety of scenarios. In each scenario, we explain the method to optimize $L_{\text{sim}}$ and the retention of task-relevant information, the experimental setup, and results. All code is at `https://tinyurl.com/simple-bits`.

## 3 PER-INSTANCE SIMPLIFICATION DURING TRAINING

When plugged into the training of a classifier $f$, *SimpleBits* simplifies each image such that $f$ can still learn the original classification task from the simplified batches. We apply backpropagation through training steps: given a batch of input images $\boldsymbol{X}_{\text{orig}}$, before updating the classifier $f$, an image-to-image simplifier network generates a corresponding batch of images $\boldsymbol{X}_{\text{sim}}$ such that: **(a)** images in $\boldsymbol{X}_{\text{sim}}$ have low bpd as measured per $L_{\text{sim}}$ in Equation (1), and **(b)** training on the simplified images leads to a reduction of the classification loss on the original images.

We optimize **(b)** by unrolling one training step of the classifier. So for a single batch $\boldsymbol{X}_{\text{orig}}, \boldsymbol{y}$, we first compute an updated classifier $f'$ as follows:

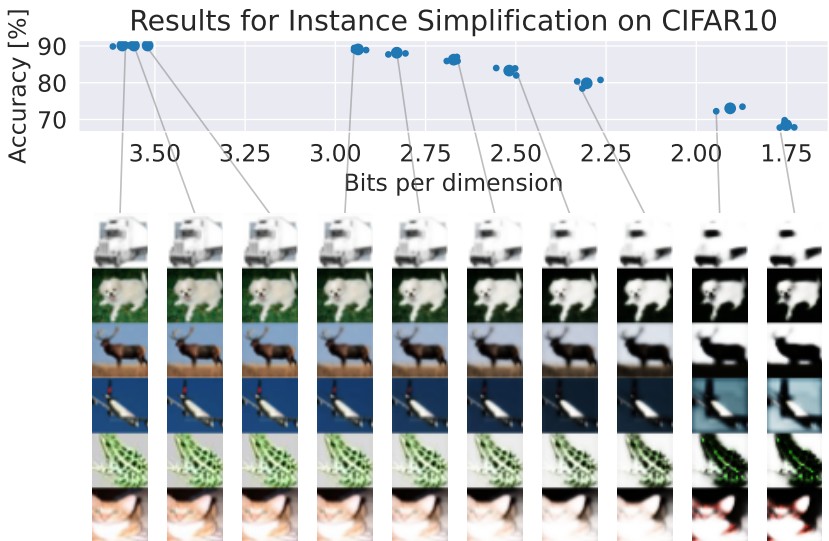

Figure 4: Selection of Simplified Training Images on CIFAR10. Simplified images from settings with varying simplification loss weight $\lambda_{\text{sim}}$. We observe that at lower bits per dimension color is retained only for some images such as green color for frog or blue sky for the plane. In this low bit regime, texture remains discernable for the cat and frog images.

$$\boldsymbol{X}_{\text{sim}} = \text{simplifier}(\boldsymbol{X}_{\text{orig}}) \tag{2}$$

$$f' = \text{train\_step}(f, l(f(\boldsymbol{X}_{\text{sim}}), \boldsymbol{y})) \tag{3}$$

where simplifier is an image-to-image network, $l$ is the classification loss function, i.e., the cross-entropy loss, and $f'$ is the classifier $f$ after one optimization step using the classification loss $l(f(\boldsymbol{X}_{\text{sim}}), \boldsymbol{y})$ on the simplified data.

To train the simplifier network, we optimize for both **(a)** (Equation (1)) and for the classification loss on the batch of original images with the updated classifier after the unrolled training step $l(f'(\boldsymbol{X}_{\text{orig}}), \boldsymbol{y})$ using backpropagation through training (Maclaurin et al., 2015; Finn et al., 2017). Note that it would not be enough to instead optimize performance on the simplified images as the simplifier could then to collapse all simplified images of one class to one representative example.

Adding the classification losses on the simplified data both before and after the unrolled training step ensures the training is not influenced by predictions on the simplified data that are very different from the classification target and we found that to improve training stability, leading to:

$$L_{\text{cls}} = l(f'(\boldsymbol{X}_{\text{orig}}), \boldsymbol{y}) + l(f(\boldsymbol{X}_{\text{sim}}), \boldsymbol{y}) + l(f'(\boldsymbol{X}_{\text{sim}}), \boldsymbol{y}) \tag{4}$$

Further details to stabilize the training are described in S4. The total loss for the simplifier is

$$L = \lambda_{\text{sim}} \cdot L_{\text{sim}} + L_{cls}, \tag{5}$$

where $\lambda_{\text{sim}}$ is a hyperparameter to control the trade-off between simplification and task completion. In subsequent experiments, we vary $\lambda_{\text{sim}}$ to flexibly control the extent of simplification.

**Implementation** Our training architecture is a normalizer-free classifier architecture to avoid interpretation difficulties that may arise from normalization layers, such as image values being downscaled by the simplifier and then renormalized again. We use Wide Residual Networks and adapt them according to the steps outlined in (Brock et al., 2021); additional details are included in the Supp. Section S2. The normalizer-free architecture reaches 94.0% on CIFAR10 in our experiments, however we opt for a smaller variant for faster experiments that reaches 91.2%, see Supp. Section S2. For the simplifier network, we use a UNet architecture (Ronneberger et al., 2015) that we modify to be residual, more details see Supp. Section S3. For both simplifier and classifier networks, we use AdamW (Loshchilov & Hutter, 2019) with $\text{lr} = 5 \cdot 10^{-4}$ and $\text{weight\_decay} = 10^{-5}$.

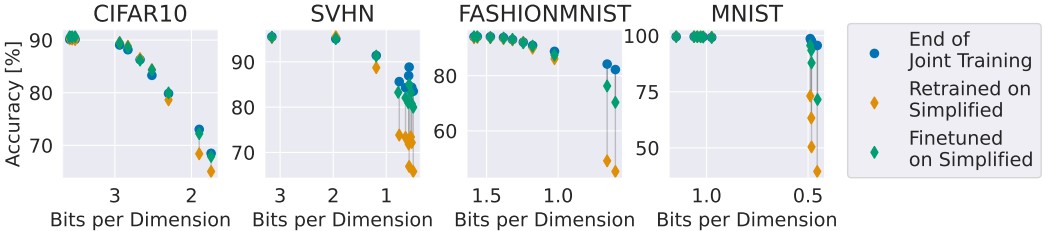

Figure 5: Results for Training Image Simplifications on real Datasets. Dots show results for training with different loss weights for the simplification loss. Images with less bits per dimension lead to reduced accuracies, this already happens for smaller bpd-reductions for more complex datasets like CIFAR10 than for less complex ones like SVHN.

### 3.1 *SimpleBits* REMOVES INJECTED DISTRACTORS

We first evaluate whether our per-instance simplification during training successfully removes superfluous information for tasks with injected distractors. To that end, we construct datasets to contain both useful (ground truth) and redundant (distractor) information for task learning. We create four composite datasets derived from three conventional datasets: MNIST (LeCun & Cortes, 2010), FashionMNIST (Xiao et al., 2017) and CIFAR10 (Krizhevsky, 2009). Their construction, samples and results are described below, and shown in Figure 3.

**Side-by-Side MNISTS** constructs each image by concatenating, left and right, one sample from Fashion-MNIST and another from MNIST. Each sample is rescaled to 16x32, so the concatenated image size remains 32x32; the order of concatenation is random. The ground truth target is MNIST labels, and therefore FashionMNIST acts like a distractor as it is irrelevant for the classification task. As seen in Figure 3, the simplifier effectively removes the clothes side of the image.

**MNIST with Uniform Noise** adds uniform noise to the MNIST digits, preserving the MNIST digit as the classification target. Hence the noise is the distractor and is expected to be removed. And indeed the noise is no longer visible in the simplified outputs shown in Figure 3.

**Interpolated MNIST and CIFAR10** is constructed by interpolating between MNIST and CIFAR10 images. MNIST digits are the classification target. The expectation is that the simplified images should no longer contain any of the CIFAR10 image information. The result shows that most of the CIFAR10 background is removed, leaving only slight traces of colors.

**CIFAR10 with Stripes** overlays either horizontal or vertical stripes onto CIFAR10 images, with the binary classification label 0 for horizontal and 1 for vertical stripes. With this dataset we observe the most drastic and effective information removal, where only the tip of vertical strips is retained, which by itself is sufficient to solve this binary classification task.

### 3.2 TRADE-OFF CURVES ON CONVENTIONAL DATASETS

Sec 3.1 evaluates the ability of *SimpleBits* to remove redundant information where ground truth is known. Now, we evaluate the trade-off between task performance and input simplification on real-world datasets. We perform instance-level simplification on MNIST, Fashion-MNIST, SVHN (Netzer et al., 2011) and CIFAR10, comparing results with varying weight $\lambda_{\text{sim}}$ for the simplification loss, including $\lambda_{\text{sim}} = 0$, so without using any simplification loss. At an extreme, when we have simplified to the black image seen in Figure 2, we expect to see significant erosion of performance. Understanding the trade-offs inbetween the original image and this extreme can help understand how sensitive the training is to the removal of more complex features.

In Figure 5, there is a pronounced decay in performance at high levels of simplify (where final bpd is less than 2). We also observe this decay curve to be more pronounced for relatively more complex datasets such as CIFAR10, suggesting the classifiers need to be trained with more complex features present for highest task performance. Baselines for our simplifier can be found in S8.

The visible retention of task-relevant information in some images together with the decreased accuracies suggests another factor: The simplified inputs may lack noise-features that the classifier has to learn to be invariant to. This is analogous to how removing data augmentation can decrease accuracies even though the augmentations themselves do not contain discriminative information. This view implies *SimpleBits* can still be used to visualize task-relevant information and at the same motivates the adaptation of *SimpleBits* to analyze regularly trained classifiers after training in Section 5.

## 4 DATASET SIMPLIFICATION WITH CONDENSATION

Now we investigate how *SimpleBits* affects training on a small synthetic condensed dataset. Multiple methods have been proposed to achieve dataset condensation (Zhao et al., 2021; Zhao & Bilen, 2021; Wang et al., 2018; Maclaurin et al., 2015), via backpropagation through training (Wang et al., 2018; Maclaurin et al., 2015), gradient matching (Zhao & Bilen, 2021), or kernel based meta-learning (Nguyen et al., 2021). Due to its small size, one can visualize the full condensed dataset to understand what information is preserved for learning. Our aim here is to combine *SimpleBits* with dataset condensation to see if we could obtain a both smaller and simpler training dataset than the original.

In this setting, we jointly condense our training dataset to a smaller number of synthetic training inputs and simplify the synthetic inputs according to our simplification loss (Equation (1)). Concretely, we add the simplification loss $L_{\text{sim}}$ to the gradient matching loss proposed by Zhao & Bilen (2021). The gradient matching loss computes the layerwise cosine distance between the gradient of the classification loss wrt. to the classifier parameters $\theta$ produced by a batch of original images $\boldsymbol{X}_{\text{orig}}$ and a batch of synthetic images $\boldsymbol{X}_{\text{syn}}$:

$$L_{\text{match}}(\boldsymbol{X}_{\text{orig}}, \boldsymbol{X}_{\text{syn}}) = D(\nabla_\theta l(f(\boldsymbol{X}_{\text{orig}}), \boldsymbol{y}), \nabla_\theta l(f(\boldsymbol{X}_{\text{syn}}), \boldsymbol{y})). \tag{6}$$

where $D$ is the layerwise cosine distance. The matching loss is computed separately per class.

Overall, with our simplification loss, we get:

$$L_{\text{syn}}(\boldsymbol{X}_{\text{orig}}, \boldsymbol{X}_{\text{syn}}) = L_{\text{match}}(\boldsymbol{X}_{\text{orig}}, \boldsymbol{X}_{\text{syn}}) + \sum_{\boldsymbol{x}_{\text{syn}} \in \boldsymbol{X}_{\text{syn}}} -\log p_G(\boldsymbol{x}_{\text{syn}}) \tag{7}$$

We perform dataset condensation on MNIST, Fashion-MNIST, SVHN and CIFAR10 with varying $\lambda_{\text{sim}}$ for the simplification loss. We also apply dataset condensation to the chest radiograph dataset MIMIC-CXR-JPG (Johnson et al., 2019a;b) for predicting pleural effusion and gender. We use the networks from (Zhao et al., 2021), but use Adam (Kingma & Ba, 2015) for optimization.

### 4.1 *SimpleBits* RETAINS CONDENSATION PERFORMANCE WHILE GREATLY SIMPLIFYING DATA

In Figure 6, we examine the accuracy for each condensed-and-simplified dataset. We observe that for the natural image datasets, accuracies are mostly retained when decreasing the number of bits per image. Note that the setting with highest bpd is a reimplementation of Zhao et al. (2021) and therefore a baseline without simplification loss. We visualize examples in Figure 6 and observe that the jointly condensed and simplified images look visually smoother, indicating that higher frequency patterns visible in the original images are not needed to reach the same accuracy. These visualizations are also noticeably more smooth than the results for per-instance simplification Figure 3, which suggests that data condensation may already favor features that are less complex.

**Evaluation of a medical chest radiograph dataset** We also evaluate jointly condensing and simplifying for a dataset of chest radiograph images (Johnson et al., 2019a;b). This dataset has known radiologic features for the presence of pleural effusion (Jany & Welte, 2019; Raasch et al., 1982) and difference in gender (Bellemare et al., 2003). In Figure 7, we visualize both the condensed **(top row)** and the jointly condensed and simplified dataset **(bottom row)**. The overlayed shows that a visible difference between presence of feature. For pleural effusion, a larger white region on the bottom of the lung occurs in the simplified pathological image, while for gender, lungs appear slightly smaller for the simplified female image.

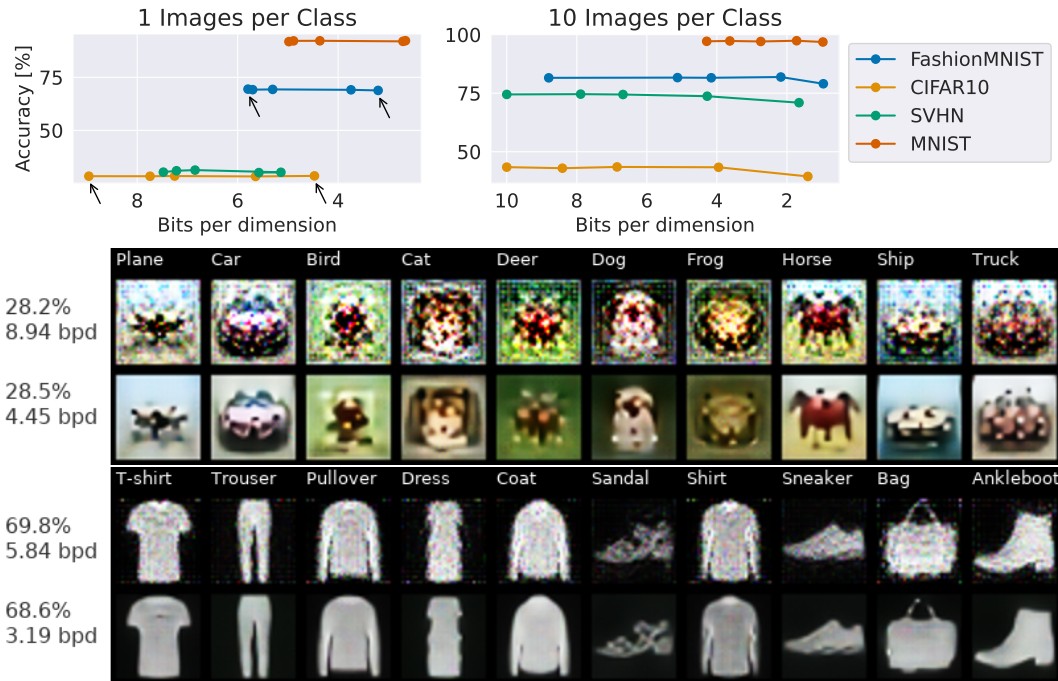

Figure 6: Dataset condensation accuracies (when retraining with the condensed dataset) vs. data simplicity. **Top:** Each dot represents a data condensation experiment run with a particular weight for the simplification loss, which results in more or less complex datasets. Accuracies can be retained even with substantially reduced bits per dimension. In the 1-image-per-class case (**top left figure**), arrows highlight the settings that are visualized in the bottom figure. **Bottom:** Condensed datasets with varying simplification loss weight. Each row represents the whole condensed dataset (1 image per class), with high (top row) or low (bottom row) bits per dimension. Lower bits per dimension datasets are visually simpler and smoother while retaining the accuracy.

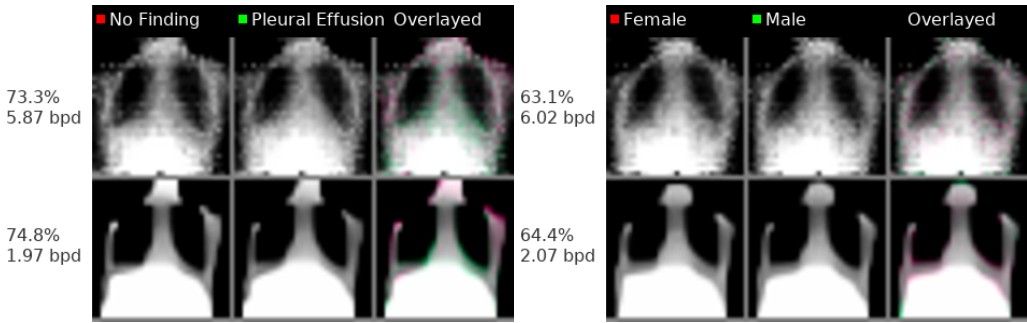

Figure 7: Condensed dataset for pleural effusion and gender prediction from chest radiographs in MIMIC-CXR. Condensed images for the classes look very similar. Color-coded mixed rightmost images reveal the differences between the classes. Green highlighted region at the lower end of the lung consistent with typical radiologic features for pleural effusion (white region indicating fluid on lungs), red highlighted around lung for gender indicate smaller lung volume for the female class.

## 5 POST-TRAINING SIMPLIFICATION

Per-instance *SimpleBits* reduces accuracies when applied during training, but can it be used to interpret trained classifiers with high accuracies *after* training? A trained classifier's prediction may be influenced by a lot of information it has seen during training. Here, we use *SimpleBits* to visualize some of the information that would help the classifier remember what it has learned for that specific prediction.

---

**Algorithm 1** Simplification loss function after training

1: **given** generative network $G$, input $\boldsymbol{x}$, simplified input $\boldsymbol{x_{sim}}$, classifier $f$, parameter scaling factors $\boldsymbol{s} < 1$
2: $f_{scd} \leftarrow$ ScaleParameters$(f, \boldsymbol{s})$  ▷ Scale parameters of classifier down by $s$ to simulate "forgetting"
3: $\boldsymbol{h} = f(\boldsymbol{x}), \boldsymbol{h}_{sim} = f(\boldsymbol{x}_{sim})$  ▷ Predict original and simplified with unscaled classifier
4: $\boldsymbol{h}_{scd} = f_{scd}(\boldsymbol{x}), \boldsymbol{h}_{sim,scd} = f_{scd}(\boldsymbol{x}_{sim})$  ▷ Predict original and simplified with scaled classifier
5: $L_{grad} = D\big(\nabla_{\boldsymbol{s}} D_{KL}(\boldsymbol{h}\|\boldsymbol{h}_{scd}), \nabla_{\boldsymbol{s}} D_{KL}(\boldsymbol{h}\|\boldsymbol{h}_{sim,scd})\big)$ ▷ Compute distance of gradients on scaling factors
6: $L_{pred} = D_{KL}(\boldsymbol{h}\|\boldsymbol{h}_{sim}) + D_{KL}(\boldsymbol{h}_{scd}\|\boldsymbol{h}_{sim,scd})$  ▷ Compute prediction differences
7: $L_{sim} = -\log p_G(\boldsymbol{x}_{sim})$  ▷ Compute needed bits for simplified input
8: **return** $L_{grad} + L_{pred} + \lambda_{sim} \cdot L_{sim}$

---

For synthesizing the prediction-relevant information, we simulate that the classifier forgets knowledge and then synthesize a simplified input that allows the classifier to relearn the relevant knowledge for the prediction of the original input. To simulate forgetting, we scale down all parameters of the trained classifier $f$ by multiplying them with a gating value $\phi_{scaled,i} = \phi_i \cdot s, s < 1$. This scaling removes information from the model by bringing the learned parameter values closer to zero, is identical to weight decay and also makes the network simpler in terms of model encoding size (Hinton & van Camp, 1993). To synthesize a simplified input that helps learning to restore the parameter values that are important for a specific input, we compare gradients between the original and simplified input.

Given an input, we can compute the gradients of the KL-divergence between the rescaled network's prediction $f_{scd}(x)$ and the original network's prediction $f(x)$. At each iteration, the layerwise cosine distance between these two sets of gradients (one for original datapoint and the other for the simplified) is the basis of the loss used in *SimpleBits*. We compute this distance only considering the gradients that are negative for the original input. Combined with the simplification loss Eq. 1, this amounts to asking *what input information is needed to recover parts of the original network to restore the original prediction?* To ensure that the network is trained towards minimizing the same prediction difference on the simplified and original data, we also add a prediction difference loss $L_{pred}$. Further details about the implementation are included in Alg. 1 and Section S5.

In Figure 8, we visualize both the misclassified images according to the original network $f$ and produce the corresponding simplified versions. We observe that simplified images may provide some intuition into the reason for misclassification, highlighting a variety of different features for different images. We imagine a possible practitioner workflow, where the practitioner derives a set of possible hypotheses for the misclassification from *SimpleBits* and tests them on the real data. We show further post-hoc simplified examples in supplementary Section S13.

## 6 RELATED WORK

A different approach to reduce input bits while retaining classification performance is to train a compressor that only keeps information that is invariant to predefined label-preserving augmentations. Dubois et al. (2021) implement this elegant approach in two ways. In their first variant, by training a VAE to reconstruct an unaugmented input from augmented (e.g. rotated, translated, sheared) versions. In their second variant, building on the CLIP (Radford et al., 2021) model, they take image-text pairs and view all images with the same text as augmented versions of each other. This allows to use compressed CLIP encodings for classification and achieves up to 1000x compression on Imagenet without decreasing classification accuracy. Their approach focuses on achieving maximum compression while our approach is focused on interpretability. Their approach requires access to predefined label-preserving augmentations and has reduced classification performance in input space compared to latent/encoding space.

Our approach simplifying individual training images builds on Raghu et al. (2021), where they learn to inject information into the classifier training. Per-instance simplification during training can be seen as a instance of their framework combined with the idea of input simplification. In difference to their methods, *SimpleBits* explicitly aims for interpretability through input simplification.

Other interpretability approaches that synthesize inputs include generating counterfactual inputs (Hvilshøj et al., 2021; Dombrowski et al., 2021; Goyal et al., 2019) or inputs with exaggerated features (Singla et al., 2020). *SimpleBits* differs in explicitly optimizing the inputs to be simpler.

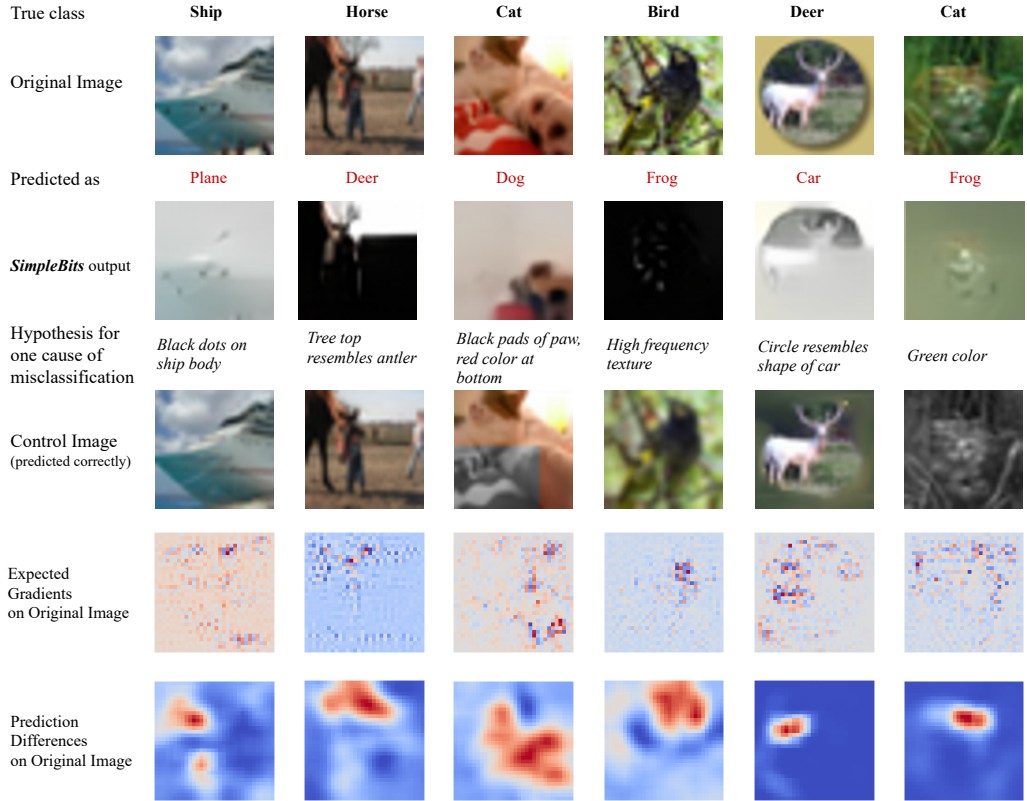

Figure 8: Post-hoc simplifications of misclassified CIFAR-10 examples. For each, simplified image reveals plausible causes for the misclassification. We subsequently made alterations to compensate for the cause (from left to right: removing black dots, removing tree top, removing color, removing high frequency texture, removing circle, and removing color), and are able to revert the predictions to the true class. We also show color-coded saliency maps for expected gradients (Erion et al., 2021) and prediction difference (Zintgraf et al., 2017) for comparison (red: evidence for and blue: evidence against the predicted class). *SimpleBits* reveals more information than saliency methods.

Generative models have often been used in various ways for interpretability such as generating realistic-looking inputs (Montavon et al., 2018) and by directly training generative classifiers (Hvilshøj et al., 2021; Dombrowski et al., 2021), but we are not aware of any work except (Dubois et al., 2021) (discussed above) to explicitly generate simpler inputs.

# 7 CONCLUSION

We propose *SimpleBits*, an information-based method to synthesize simplified inputs. Crucially, *SimpleBits* does not require any domain-specific knowledge to constrain which input components should be removed; instead *SimpleBits* itself learns to remove the components of inputs which are least relevant for a given task.

As an interpretability tool, we show that *SimpleBits* is able to remove injected distractors, suggest plausible reasons for misclassification, and recover known radiologic features from condensed datasets. When combined with data condensation, *SimpleBits* retains accuracy while greatly reducing the complexity of condensed images.

Our simplification approach sheds light on the information required for a deep network classifier to learn its task. We find that the tradeoff between task performance and input simplification varies by dataset and setting - it is more pronounced for more complex datasets.

## REPRODUCIBILITY STATEMENT

We provide the following information to ensure reproducibility. Main concepts, algorithms and basic architectures are described in sections 3, 5 and 4. Further network architecture details are in supp. sections S2 and S3, further optimization details in supp. sections S4 and S5. Finally, the code is available under `https://tinyurl.com/simple-bits`.

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

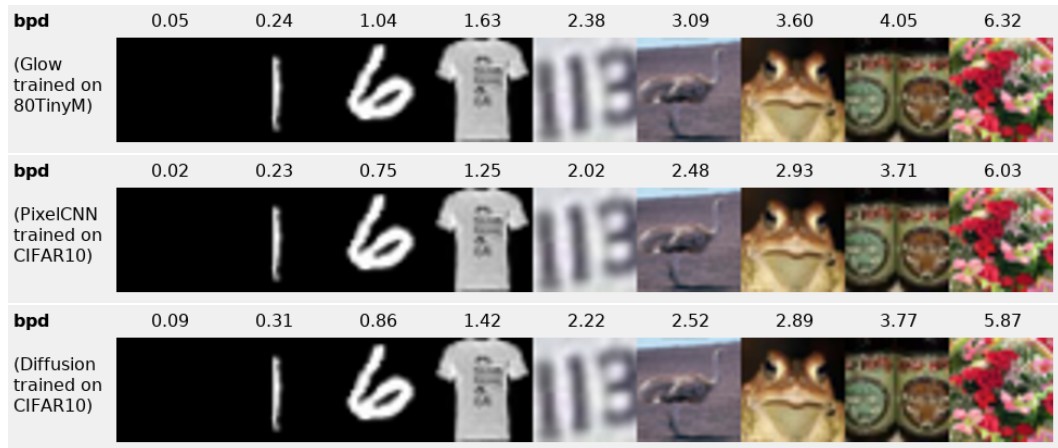

Figure S1: Visualization of the bits-per-dimension (bpd) measure for image complexity, sorted from low to high. Image samples are taken from MNIST, Fashion-MNIST, CIFAR10 and CIFAR100, in addition to a completely black image sample. bpd is calculated from the density produced by a Glow (Kingma & Dhariwal, 2018) model pretrained on 80 Million Tiny Images, a PixelCNN model trained on CIFAR10, and a diffusion model trained on CIFAR10.

## SUPPLEMENTARY OUTLINE

This document completes the presentation of the main paper with the following:

- Bits per dimensions for selected images of other generative models in S1
- Details about the classifier and simplifier architectures in sections S2 and S3
- Details about the optimization of per-instance simplification during and after training in sections S4 and S5
- Files sizes of the simplified images when compressed with PNG in section S6
- Learning curves during retraining on the simplified images in section S7
- Baselines for the per-instance simplification during training in section S8
- Potentially spurious features revealed by *SimpleBits* in section S10
- More images simplified during and after training in sections S9 and S13, including post-hoc-*SimpleBits* output on the control images
- More condensed and simplified datasets under varying settings in section S11
- Results for the continual learning setting in section S12

## S1 BPDs OF OTHER GENERATIVE MODELS

Figure S1 shows that the bits per dimension produced by other generative models than Glow also correlate well with visual complexity, validating our measure. This is consistent with prior work that found bpds of generative models trained on natural image datasets are strongly influenced by general natural image characteristics independent of any specific dataset (Kirichenko et al., 2020; Schirrmeister et al., 2020; Havtorn et al., 2021).

## S2 CLASSIFIER NETWORK DETAILS

Our classification network built on the Wide ResNet architecture (Zagoruyko & Komodakis, 2016). We used a version with relatively few parameters with depth = 16 and widen_factor = 2 to allow for fast iteration on experimentation. We used ELU instead of ReLU nonlinearities.

Additionally, we removed batch normalization to avoid interference of normalization layers with the simplification process. We followed the method from Brock et al. (2021) to create a normalizer-free Wide ResNet. We reparameterize the convolutional layers using Scaled Weight Standardization:

$$\hat{W}_{ij} = \frac{W_{ij} - \mu_i}{\sqrt{N}\sigma_i}, \tag{S1}$$

where $\mu_i = (1/N)\sum_j W_{ij}$, $\sigma_i^2 = (1/N)\sum_j (W_{ij} - \mu_i)^2$, and $N$ denotes the fan-in. Further as in Brock et al. (2021), "activation functions are also scaled by a non-linearity specific scalar gain $\gamma$, which ensures that the combination of the $\gamma$-scaled activation function and a Scaled Weight Standardized layer is variance preserving." Finally, the output of the residual branch is downscaled by 0.2, so the function to compute the output becomes $h_{i+1} = h_i + 0.2 \cdot f_i(h_i)$, where $h_i$ denotes the inputs to the $i^{th}$ residual block, and $f_i$ denotes the function computed by the $i^{th}$ residual branch. Unlike Brock et al. (2021), we did not multiply scalars $\beta_i$ with the input of the residual branch or learned zero-initialized scalars to multiply with the output of the residual branch, as we did not find these two parts helpful in our setting. We also did not attempt to use Stochastic Depth (Huang et al., 2016), which may further improve upon the accuracies reported here. Due to our small batch sizes (32), we also did not use adaptive gradient clipping.

## S3  SIMPLIFIER NETWORK DETAILS

For the simplifier, we adapted a publicly available implementation of UNet [1]. We used num_down $= 5$ downsampling steps, ELU nonlinearities ngf $= 64$ filters in the last conv layer and a simple affine transformation layer instead of a normalization layer. Furthermore, we made the simplifier residual and ensured the output is within $[0, 1]$ by adding the output of the UNet to the inverse-sigmoid-transformed input and then reapplying the sigmoid function.

## S4  OPTIMIZATION DETAILS PER-INSTANCE SIMPLIFICATION DURING TRAINING

First, we note that the single train_step helps ensure a correspondence between simplified and original images, and is a technique others have used in meta-learning settings (Pham et al., 2021).

For stabilizing the optimization of the per-instance simplification during training, we found two further steps helpful. First, we modify:

$$L_{\text{cls}} = l(f'(\mathbf{X}_{\text{orig}}), \mathbf{y}) + l(f(\mathbf{X}_{\text{sim}}), \mathbf{y}) + l(f'(\mathbf{X}_{\text{sim}}), \mathbf{y}) \tag{S2}$$

to

$$L_{\text{cls}} = 10 \cdot l(f'(\mathbf{X}_{\text{orig}}), \mathbf{y}) + l(f(\mathbf{X}_{\text{sim}}), \mathbf{y}) + l(f'(\mathbf{X}_{\text{sim}}), \mathbf{y}) \tag{S3}$$

as (a) the gradient magnitudes are much smaller from the losses after unrolling and (b) we want to prioritize the classification loss on the original data. Additionally, we dynamically turn off $L_{\text{sim}}$ during training, for any example $\mathbf{x}$ where $l(f'(\mathbf{x}_{\text{orig}}), \mathbf{y}) > 0.1$, which we also found to further stabilize training.

## S5  OPTIMIZATION DETAILS PER-INSTANCE SIMPLIFICATION AFTER TRAINING

We found it beneficial to optimize the simplified inputs in the latent space of the pre-trained Glow network and to apply our loss functions to all interpolated inputs on the path between original and simple input in latent space instead of only to the simplified input itself. The points on the path include information from the original input and prevent that the optimization is unable to recover some relevant information from the original input. Additionally, we also ensure that the predictions are the same for the original and interpolated inputs for both the original and the scaled model. The complete loss function can be found in Alg. 2, during training we called it with scaling factors sampled from $s$ $U(0.8, 0.95)$ as these values mostly led to similar but more uniformly distributed predictions than the unperturbed network.

---

[1] https://github.com/junyanz/pytorch-CycleGAN-and-pix2pix

---

**Algorithm 2** Simplification loss function after training full algorithm

1: **given** generative invertible network $G$, input $\boldsymbol{x}$ and its latent code $\boldsymbol{z}$ (from $G$), simplified input's latent code $\boldsymbol{z}_{\text{sim}}$, classification network $f$, parameter scaling factors $\boldsymbol{s} < 1$
2: $f_{\text{scaled}} \leftarrow \text{ScaleParameters}(f, \boldsymbol{s})$     ▷ Scale parameters of classifier down by $\boldsymbol{s}$ to simulate "forgetting"
3: $\alpha \sim U(0, 1)$     ▷ Sample interpolation factor uniformly between 0 and 1.
4: $\boldsymbol{z}_{\text{mixed}} = \alpha \cdot \boldsymbol{z}_{sim} + (1 - \alpha) \cdot \boldsymbol{z}$     ▷ Interpolate simple and original input in latent space
5: $\boldsymbol{x}_{\text{mix}} = \text{invert}(G, \boldsymbol{z}_{\text{mix}})$     ▷ Invert $\boldsymbol{z}_{\text{mix}}$ using invertible network
6: $\boldsymbol{h} = f(\boldsymbol{x}), \boldsymbol{h}_{\text{mix}} = f(x_{\text{mix}})$     ▷ Predict original and mixed with classifier
7: $\boldsymbol{h}_{\text{scaled}} = f_{\text{scaled}}(\boldsymbol{x}), \boldsymbol{h}_{\text{scaled,mix}} = f_{\text{scaled}}(\boldsymbol{x}_{\text{mix}})$     ▷ Predict original and mixed with scaled classifier
8: $L_{\text{grad}} = d\big(\nabla_{\boldsymbol{s}} D_{\text{KL}}(\boldsymbol{h}\|\boldsymbol{h}_{\text{scaled}}), \nabla_{\boldsymbol{s}} D_{\text{KL}}(\boldsymbol{h}\|\boldsymbol{h}_{\text{mix,scaled}})\big)$     ▷ Compute distance between gradients on scaling factors.
9: $L_{\text{pred}} = D_{\text{KL}}(\boldsymbol{h}\|\boldsymbol{h}_{\text{mix}}) + D_{\text{KL}}(\boldsymbol{h}_{\text{scaled}}\|\boldsymbol{h}_{\text{mix,scaled}})$     ▷ Compute prediction differences
10: $\boldsymbol{x}_{\text{sim}} = \text{invert}(G, \boldsymbol{z}_{\text{sim}})$     ▷ Invert $\boldsymbol{z}_{\text{sim}}$ using invertible network
11: $L_{\text{simplification}} = -\log p_G(\boldsymbol{x}_{\text{sim}})$     ▷ Compute needed bits for simplified input
12: **return** $L_{\text{grad}} + L_{\text{pred}} + \lambda_{\text{sim}} \cdot L_{\text{simplification}}$

---

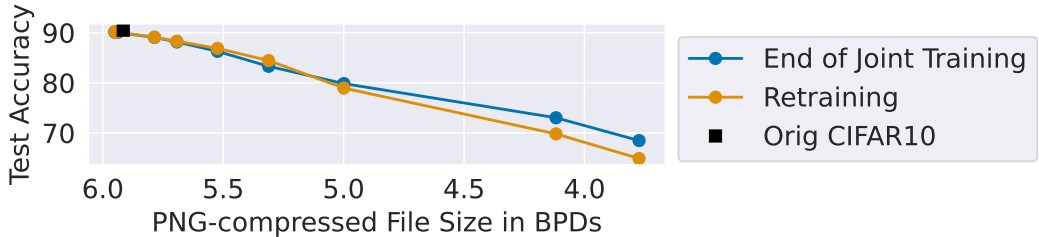

Figure S2: Tradeoff between PNG storage space and accuracies. Note that the PNG bpd file sizes do not show the maximally possible savings, these can be seen from the bpd values in the main manuscript.

## S6   PNG-COMPRESSED FILE SIZES OF SIMPLIFIED IMAGES

Figure S2 shows the tradeoff between PNG storage space of the simplified images and the accuracies achieved when retraining.

## S7   LEARNING CURVES FOR RETRAINING

Figure S3 shows learning curves during retraining on the simplified images on CIFAR10. There are no noticeable differences in training speed for more or less simplified images.

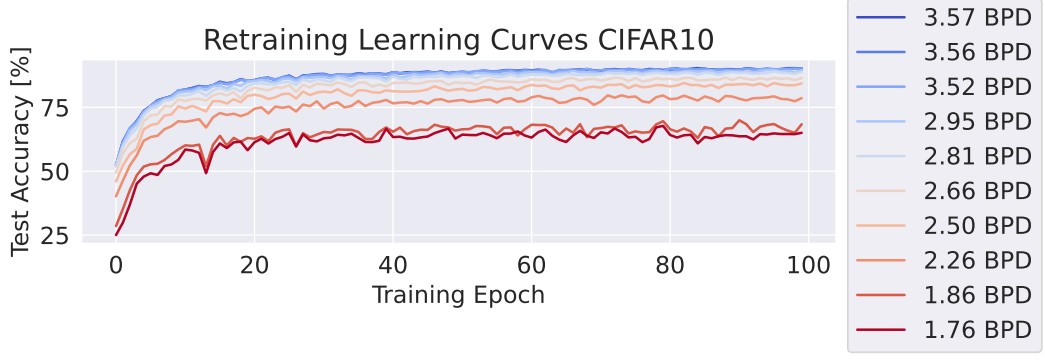

Figure S3: Learning curves for retraining on simplified images on CIFAR10.

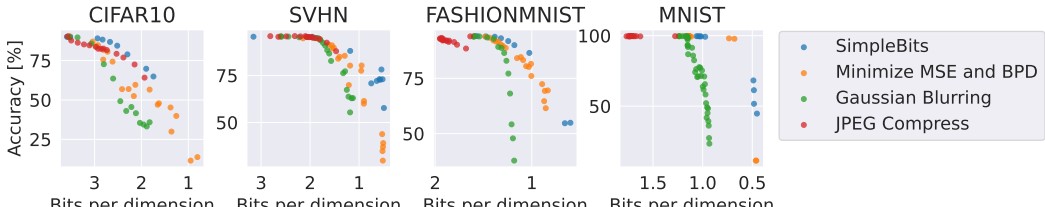

Figure S4: Comparison between *SimpleBits* and two simpler baselines: In the first one, the simplifier network is trained to simultaneously reduce bpd of the simplified image and the mean squared error between the simplified and the original image. In the second one, gaussian blurring is applied to the input images, different runs vary in the standard deviation used to create the gaussian blurring kernel. In the third one, we use JPEG compression with varying quality levels. Tradeoff curves are worse for the baselines than for *SimpleBits* .

## S8  SIMPLIFIER BASELINES

We implemented three simpler baselines to check whether the losses used in *SimpleBits* during training help retain task-relevant information. In the first baseline, we train the simplifier to simultaneously reduce bpd of the simplified image and the mean squared error between the simplified and the original image. Afterwards we train the classifier on the simplified images and evaluate on the original images the same way as during retraining of *SimpleBits*. In the second baseline, we blur the original images with a gaussian kernel, which also reduces their bpd. We vary the sigma/standard deviation for the gaussian kernel to trade off smoothness and task-informativeness. In the third baseline, we use lossy JPEG compression with varying quality levels. As in *SimpleBits* and the other baselines, we estimate the bits per dimension of the lossy-JPEG-compressed images through our pretrained Glow network for a fair comparison. The gaussian blurring and JPEG compression each replace the simplifier, so these are fixed simplifier baselines without training a simplifier. While these three baselines also retain some task-relevant information allowing the classifier to retain above-chance accuracies (see Figure S4), the tradeoff between bpd and accuracy is worse than for *SimpleBits*. This shows the losses used in *SimpleBits* help retain more task-relevant information compared to these baselines.

## S9  MORE IMAGES SIMPLIFIED DURING TRAINING

We show a larger number of images that were simplified on CIFAR10 during training with the largest simplification loss weight $\lambda_{\text{sim}} = 2.0$ in Figures S5, S6 and S7.

## S10  POTENTIALLY SPURIOUS FEATURES UNCOVERED BY SIMPLEBITS

*SimpleBits* may have the potential to reveal spurious correlations present in the dataset. We show some simplified images that reveal potentially spurious features in Figure S8. These observations can be used as a starting point to further investigate whether these features also affect regularly trained classifiers.

## S11  MORE CONDENSED DATASETS

We also show some interesting condensed datasets that resulted when we varied the architecture (Figure S10) or the condensation loss (from gradient matching to either negative gradient product or single-training-step unrolling, Figure S11).

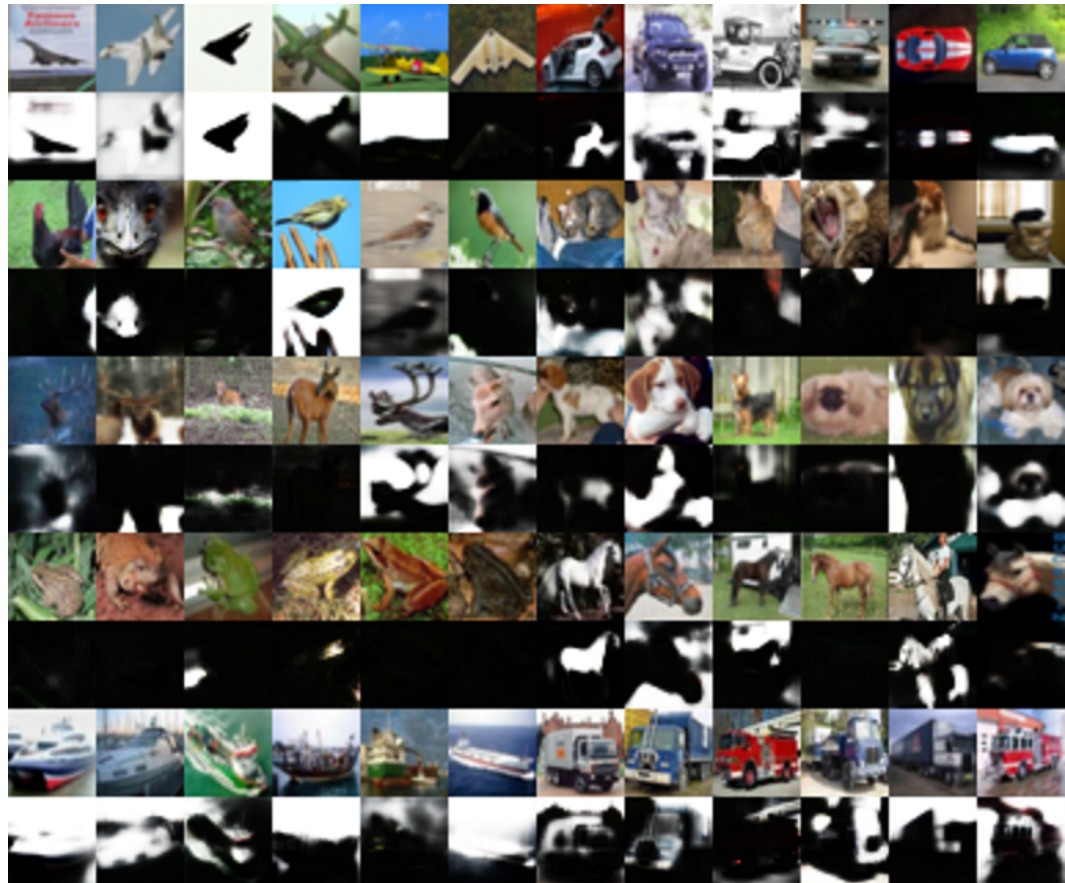

Figure S5: Uncurated set of simplified images with $\lambda_{\text{sim}} = 2.0$, 6 per class.

## S12  EVALUATION OF CONDENSED DATASETS FOR CONTINUAL LEARNING

We also evaluate the simplified condensed datasets in a continual learning setting, following (Zhao & Bilen, 2021). In this task-incremental continual learning setting, the model is trained on different classification datasets sequentially. When training on a new dataset, the model is additionally trained on the condensed versions of the previous datasets.

The continual learning experiment reproduces the setting from (Zhao & Bilen, 2021) to first train on SVHN, then on MNIST and finally on USPS (Hull, 1994), using the average accuracy across all three datasets of the classifier at the end of training as the final accuracy (see (Zhao & Bilen, 2021) for details).

We created a simpler and faster continual training pipeline that achieves comparable results to Zhao & Bilen (2021). First, we train 3 times for 50 epochs on SVHN, with a cosine annealing learning rate schedule (Loshchilov & Hutter, 2017) that is restarted at each time with $lr = 0.1$. Then for each MNIST and USPS, we train one cosine annealing cycle of 50 epochs for $lr = 0.1$.

We first verified that we can reproduce the prior continual learning results with our simpler training pipeline and find that our training pipeline indeed even slightly outperforms the reported final results (96.0% vs. 95.2% with, and 95.4% vs 93.0% without knowledge distillation) despite slightly inferior performance in the first training stage (before any continual learning, 93.6% vs 94.1%), see following subsection. When using different SVHN and MNIST condensed datasets, we find that we can retain the original continual learning accuracies even with condensed datasets with substantially less (∼9x less) bits per dimension (see Fig.S12).

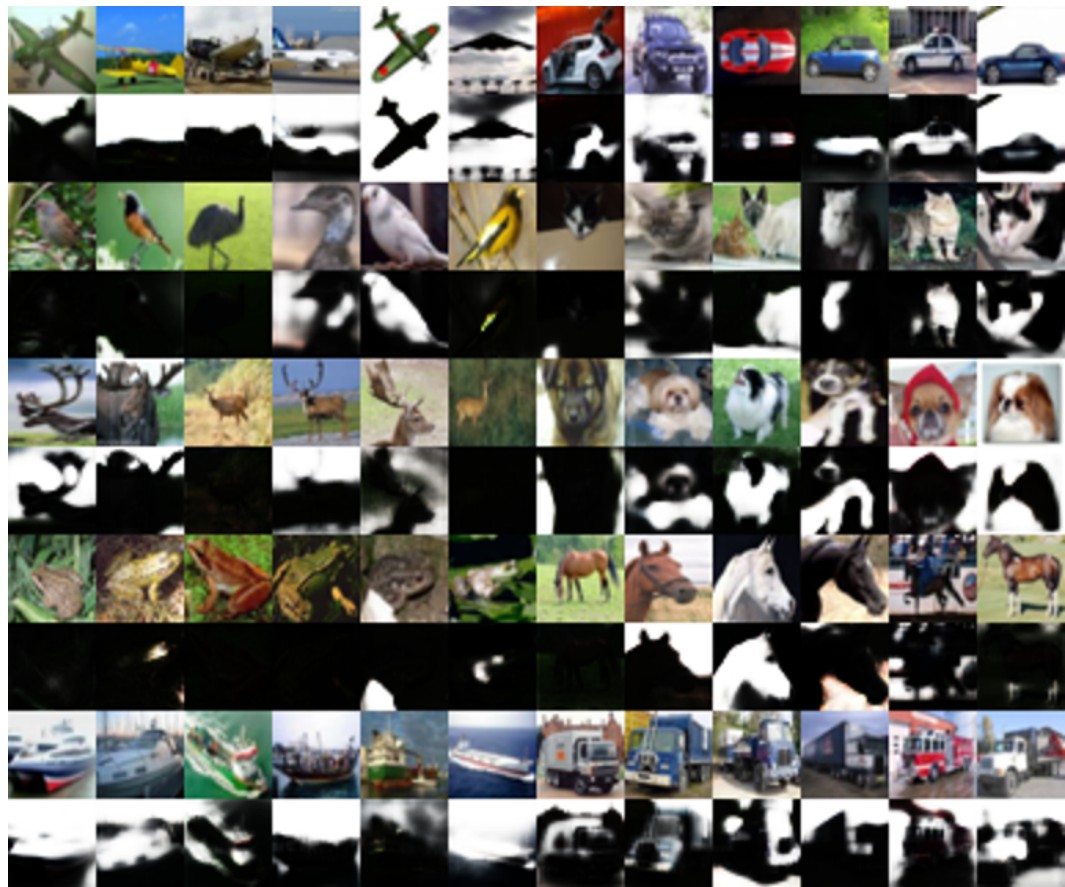

Figure S6: Uncurated set of correctly predicted simplified images with $\lambda_{sim} = 2.0$, 6 per class.

### S12.1 WITHOUT CONDENSED DATASET

Our training pipeline still exhibits forgetting when not using any condensed datasets of previously trained-on datasets. As Figure S13 shows, the accuracies are far lower than with just regular sequential training. We performed this ablation to ensure forgetting still occurs in our training pipeline.

## S13 MORE IMAGES SIMPLIFIED AFTER TRAINING

We show further examples of post-hoc-simplified images for misclassified original images in Figure S14. We also show the output of *SimpleBits* when applied to the control images of Figure 8 in Figure S15. We also show an uncurated set of incorrectly predicted and correctly predicted images in Figures S16 and S17.

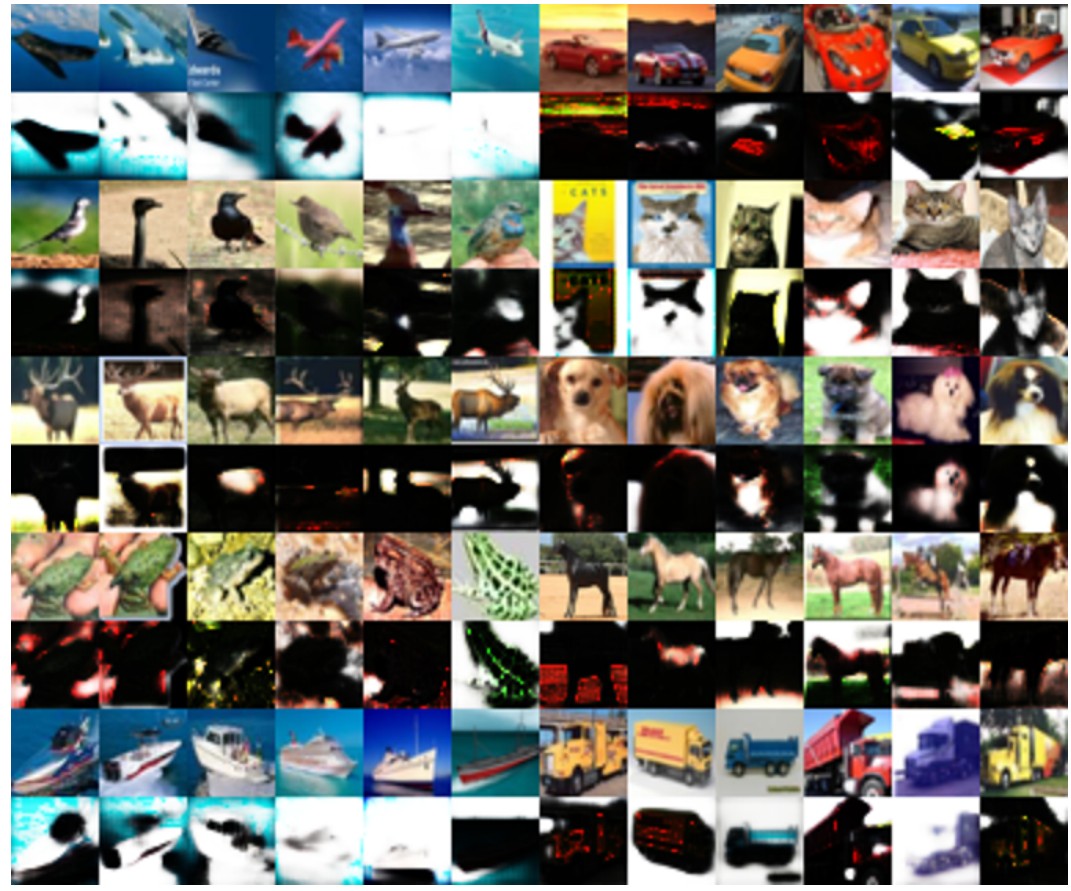

Figure S7: Correctly predicted simplified images with strongest color with $\lambda_{\text{sim}} = 2.0$, 6 per class.

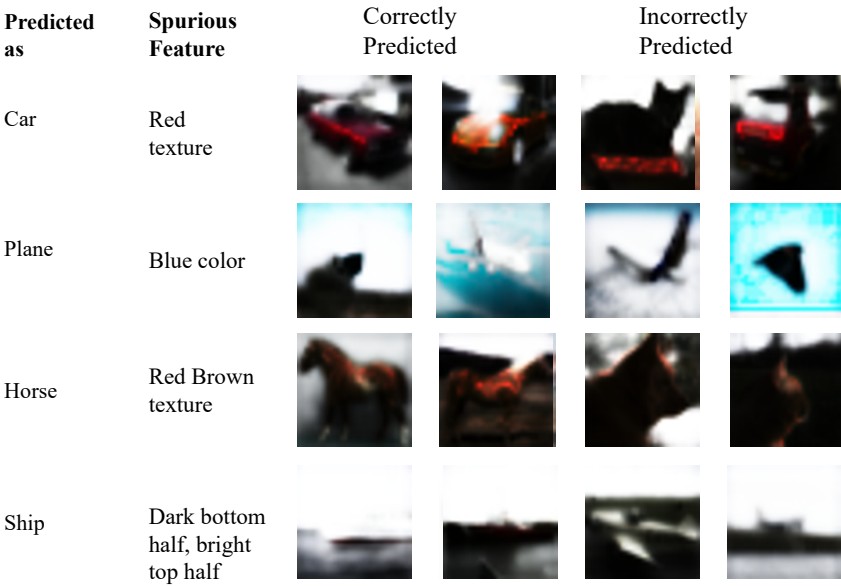

Figure S8: Selected simplified images that highlight potentially spurious features. Two leftmost images are correctly predicted, two rightmost images are incorrectly predicted.

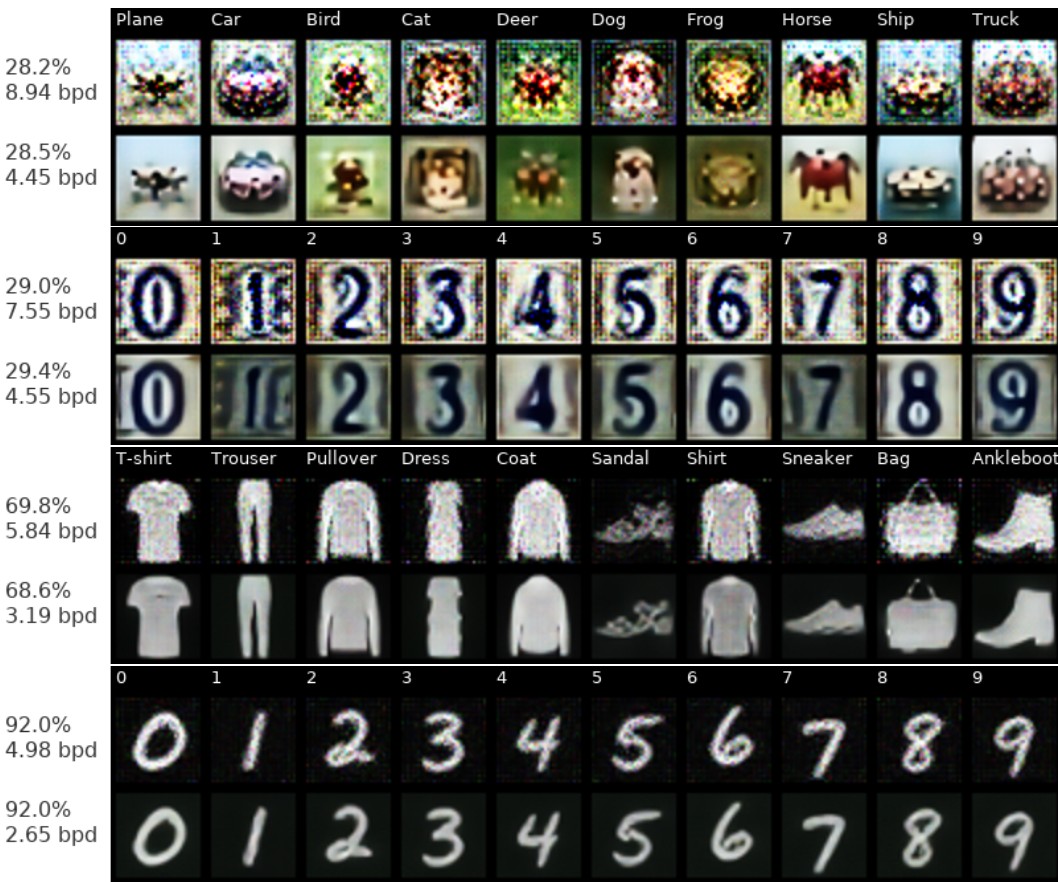

Figure S9: Dataset condensation results with varying simplification loss weight. **Top:** Individual dots represent accuracies for setting with different simplification loss weights. Accuracies can be retained even with substantially reduced bits per dimension. For 1 image per class, arrows highlight the settings that are visualized below. **Below:** Condensed datasets with varying simplification loss weight. Per dataset, showing condensed datasets with high (top row) and low (bottom row) bits per dimension. Lower bits per dimension datasets are visually simpler and smoother while mostly retaining accuracies.

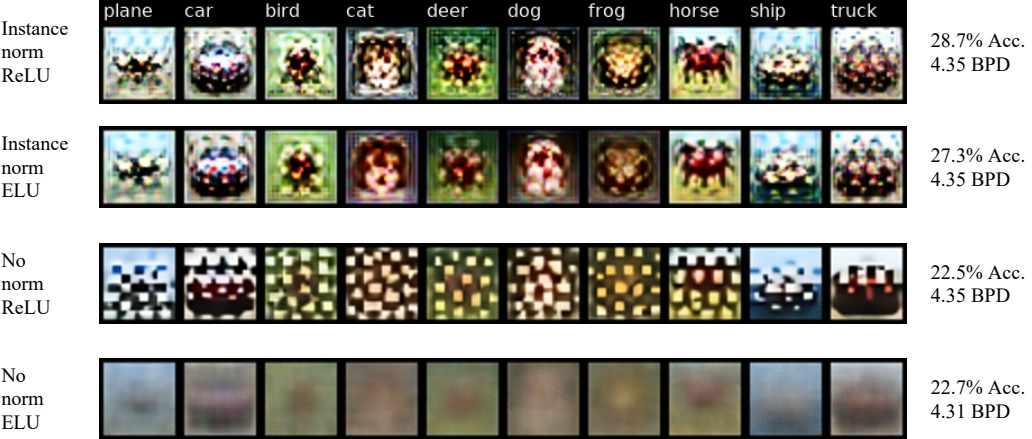

Figure S10: Dataset condensation on CIFAR10 with varying architecture.

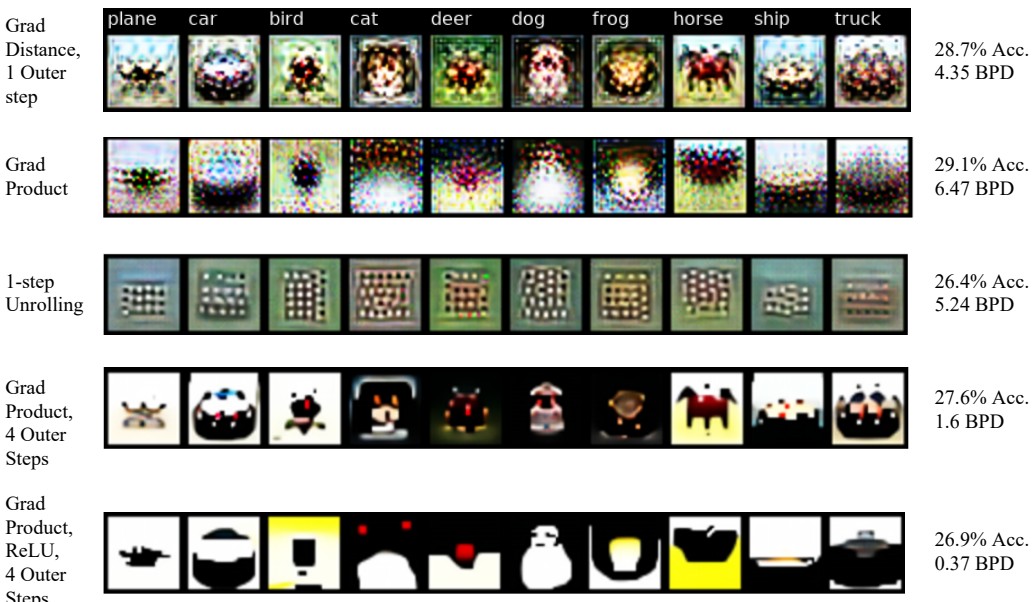

Figure S11: Dataset condensation on CIFAR10 with varying condensation loss and varying outer loop steps, i.e. how many steps the classifier is trained at each training epoch (default 1 in the 1 image per class setting), after each step the condensation loss is again optimized.

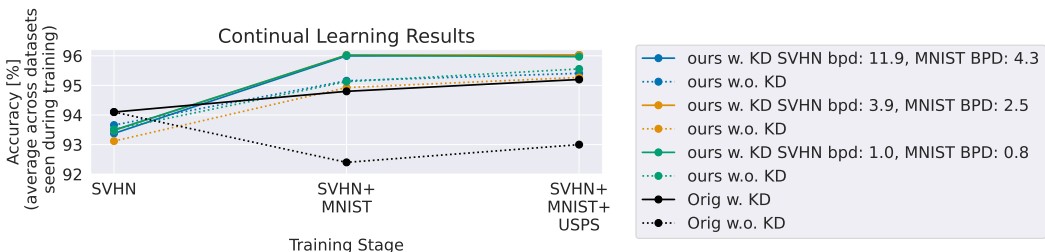

Figure S12: Continual Learning Results. Results for first training on SVHN, then MNIST and then USPS for condensed datasets with varying bits per dimension. Solid lines are with and dashed lines without knowledge distillation. Note that continual learning accuracies remain similar also for substantially reduced bits per dimension. Ablations show that accuracies degrade without any condensed dataset, see supplementary.

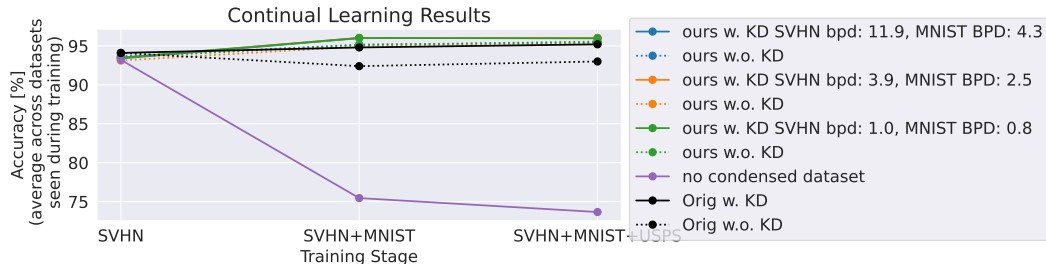

Figure S13: Continual Learning Results without Condensed Dataset (regular sequential training). Conventions as in Figure S12. Accuracies substantially worse without any condensed dataset.

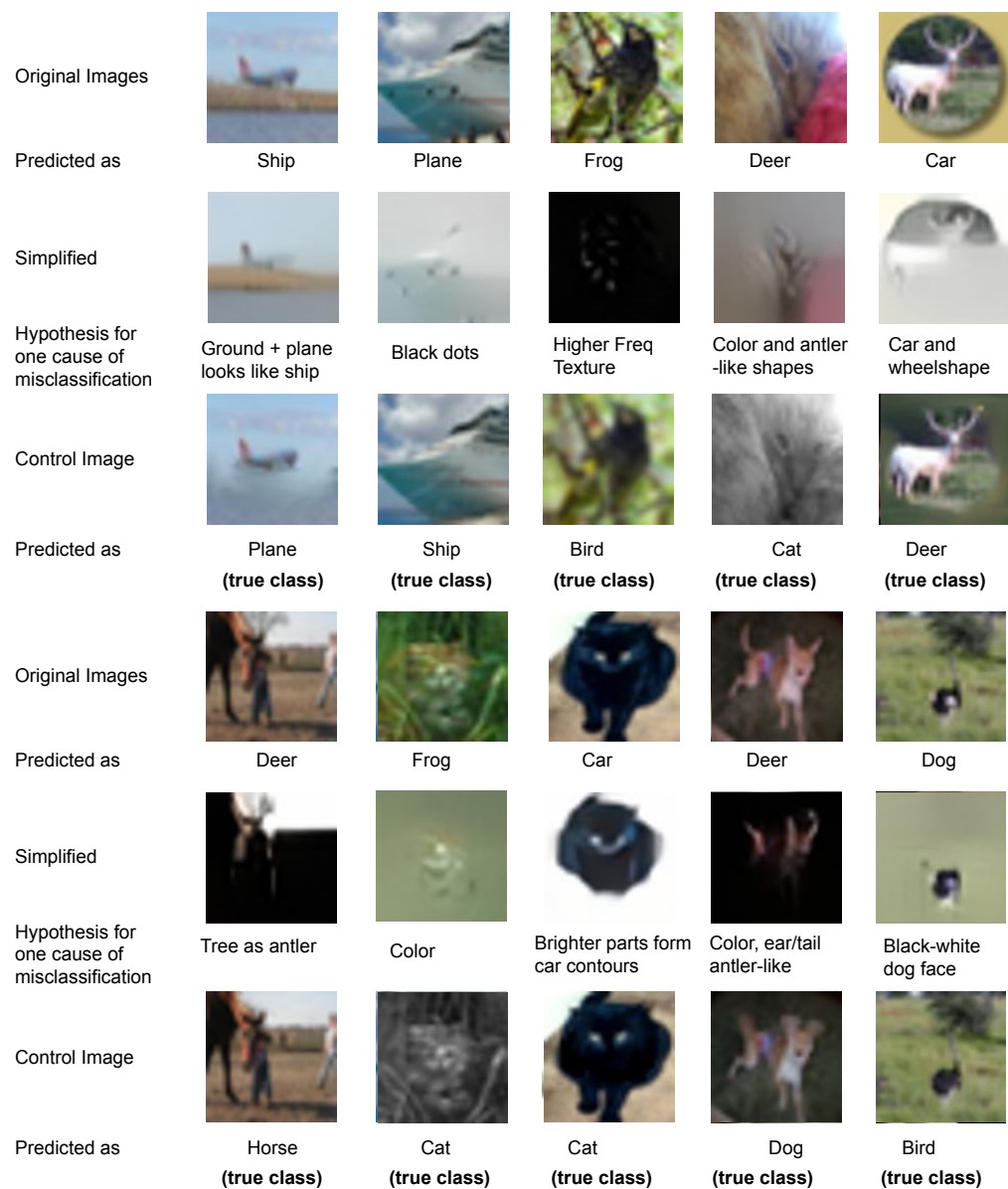

Figure S14: Further examples of post-hoc simplifications of originally misclassified images.

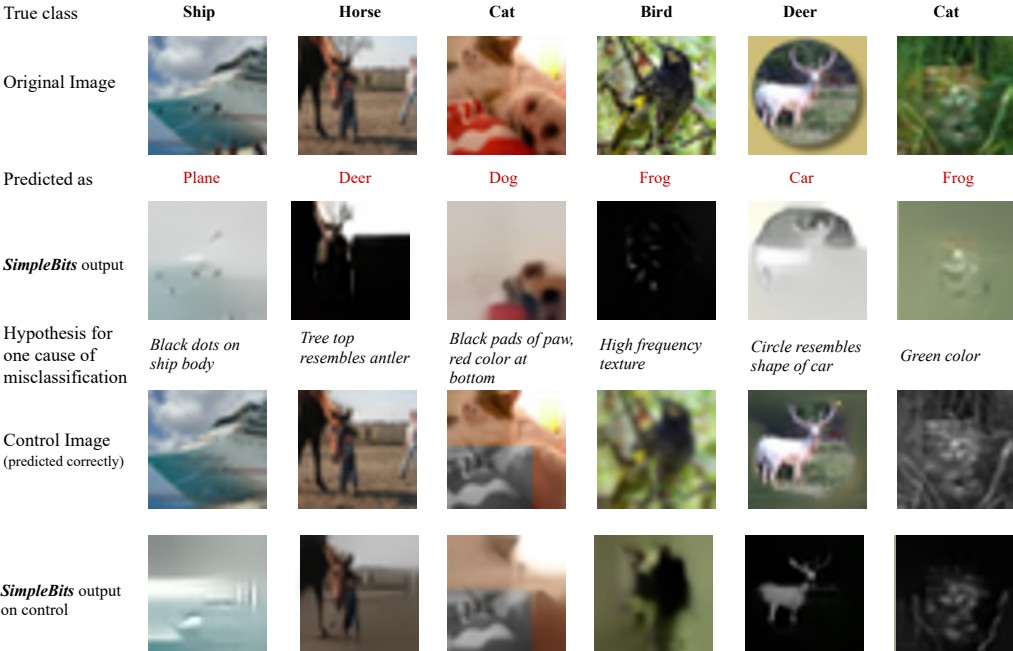

| True class | **Ship** | **Horse** | **Cat** | **Bird** | **Deer** | **Cat** |
|---|---|---|---|---|---|---|
| Original Image | | | | | | |
| Predicted as | Plane | Deer | Dog | Frog | Car | Frog |
| *SimpleBits* output | | | | | | |
| Hypothesis for one cause of misclassification | *Black dots on ship body* | *Tree top resembles antler* | *Black pads of paw, red color at bottom* | *High frequency texture* | *Circle resembles shape of car* | *Green color* |
| Control Image (predicted correctly) | | | | | | |
| *SimpleBits* output on control | | | | | | |

Figure S15: Post-hoc simplifications of control images.

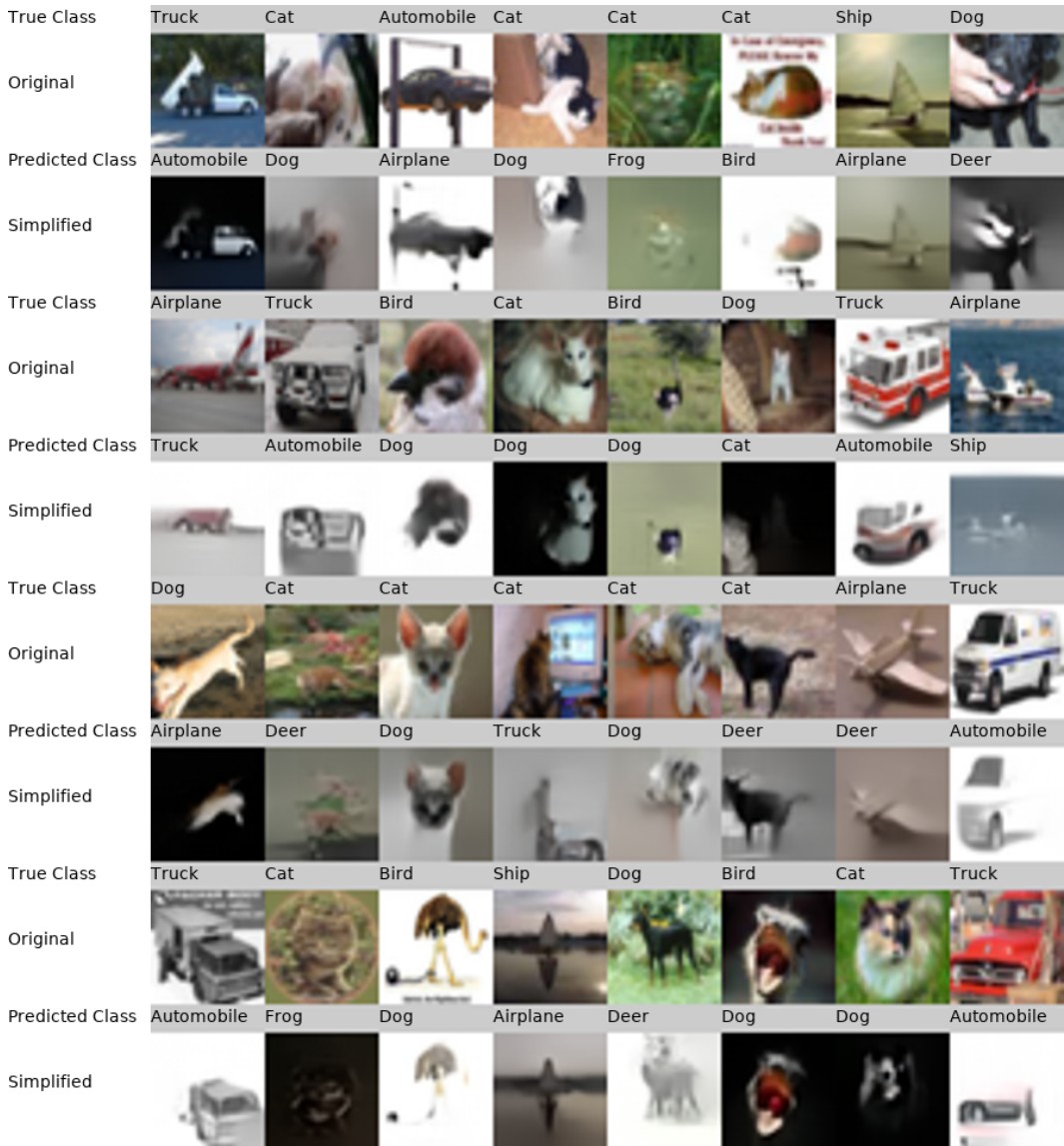

Figure S16: Uncurated post-hoc simplifications of incorrectly predicted images.

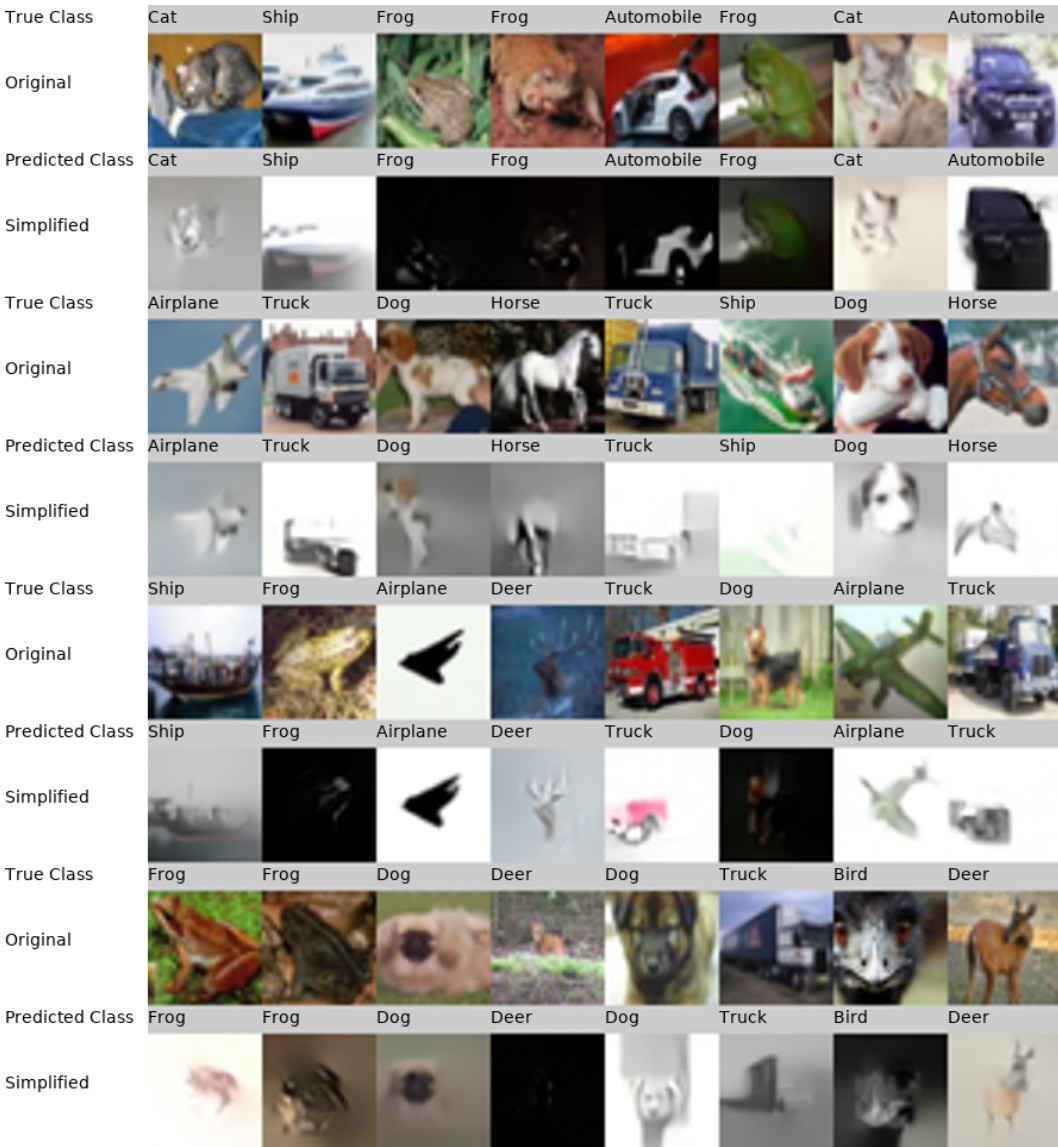

Figure S17: Uncurated post-hoc simplifications of correctly predicted images.

