# OpenReview forum: "When less is more: Simplifying inputs aids neural network understanding"
_ICLR.cc/2022/Conference — ICLR 2022 Submitted_

### Official Review · Reviewer_RfmX · 2021-10-22

**Correctness:** 3
**Technical Novelty And Significance:** 2
**Empirical Novelty And Significance:** 3
**Recommendation:** 5
**Confidence:** 4

**Main Review:**

I find the visualization tool proposed in the paper very interesting but I encountered two important limitations:

(1) Poorly explained methodology

It's not clear what the $unrolledtrainstep$ function (Eq.3) implements. I don't think this problem formalization is common knowledge, so it should be more thoroughly explained.
Similarly, the $simplifier$ function (Eq.2) is not described in the main paper.

"...adding the classification losses on the simplified data both before and after the unrolled training step improves training stability".
The rationale for *why* this works is not provided.

(2) Lack of baselines and comparisons

The simplifier function is not investigated, so it's impossible to know if the method is sensitive to this component. The choice of the specific simplifier used is not justified.

On the condensation experiments there's no comparison with other condensation methods in the literature (Fig.7).
There's also no natural baseline provided, e.g. a dataset of the same size as the original (or until performance levels off) in order to understand how good this type of condensation can be in the limit.

Similarly, the results on training with simplified images (Fig.5) do not include baselines or other competing methods.

(other comments)

Fig.6: It would be interesting to see what the SimpleBits output is for the control image. For example, does the simplified image of the deer change considerably when the background color (outside the circle) is manually corrected?
Are there cases where after correcting the suspected cause of error, the image is still misclassified? It would be good to show some unsuccessful/hard cases.

Sec.4: As a visual inspection tool, this is interesting, but the manual process of altering the images is unlikely to be viable for many applications like large datasets or streaming data). Do the authors anticipate a way of automating this process?

**Summary Of The Paper:**

This paper proposes a method to simplify images by reducing their information content, and measure the resulting effects on several aspects of learning: removing redundant image features for network training, understanding classification errors via visualization, and data condensation.

**Summary Of The Review:**

In my opinion the methodology section must be more thoroughly explained, and comparisons/baselines must be added. For these reasons I recommend that the paper is rejected in its current form. I think that the visualization tool proposed for diagnosing errors could be useful, so I would encourage the authors to improve the manuscript.

Score: 3: reject, not good enough.

============== AFTER REBUTTAL

I have updated my score for this paper, following the authors' clarifications. However, I still think that the paper is incomplete with respect to baselines using alternative image compression methods or other techniques.

Updated Score: 5: marginally below the acceptance threshold.

---

> ### Author Response · Authors · 2021-11-16
> **Thanks, comments below 1/2**
>
> Thanks for your precise questions and critiques which we try to address below.
>
> > It's not clear what the function $\mathrm{unrolledtrainstep}$ (Eq.3) implements. I don't think this problem formalization is common knowledge, so it should be more thoroughly explained. Similarly, the function $\mathrm{simplifier}$ (Eq.2) is not described in the main paper.
>
> Thank you for bringing this up. The $\mathrm{unrolledtrainstep}$ was our term for a single step of backpropagation through training as done e.g., in MAML (https://arxiv.org/abs/1703.03400), and integrated into the training process in a similar way to us in https://arxiv.org/abs/2011.03037v1  and https://arxiv.org/abs/2003.10580. Basically, you unroll the optimization update function (of e.g., ADAM or SGD) so that you can backpropagate through the training update. We have renamed it to $\mathrm{trainstep}$ as the unrolling is mentioned in the surrounding text and we have added that we use backpropagation through training with the appropriate references in the text. The simplify function is an image-to-image network, in our case a modified UNet. We have added that the simplifier is a network to the revised manuscript, thanks for the notice. We hope this section now reads more clearly, let us know if further improvements are necessary.
>
> > "...adding the classification losses on the simplified data both before and after the unrolled training step improves training stability". The rationale for why this works is not provided.
>
> Our motivation for these auxiliary losses were: 1) The gradient from the loss that the simplified image should be classified correctly before the unrolling step is unaffected by the other instances in the same batch so provides a stronger signal. 2) In general, ensuring that the prediction on the simplified image is also close to the target may prevent that the optimization leads to a simplified image that is predicted quite differently from the original image. A simplified image that is predicted quite differently may emerge if this prediction difference helps create a larger training update on the simplified image that is still beneficial for the original image at that point in training. This may lead to interpretation difficulties and we wanted to avoid that. We have added parts of this explanation to the manuscript.
>
> > The simplifier function is not investigated, so it's impossible to know if the method is sensitive to this component. The choice of the specific simplifier used is not justified.
>
> For the simplifier function, we tried to use a very common image-to-image network, as UNet is one of the most common image-to-image networks, we used it (as prior work had also done, e.g., https://arxiv.org/abs/2011.03037v1) and only modified it to be residual to speed up the training by initializing the output towards the identity function. We tried to rely on a very often-used image-to-image architecture instead of optimizing the simplifier architecture a lot, we can consider to further ablate this choice if computational resources and time permit.
>
> > On the condensation experiments there's no comparison with other condensation methods in the literature (Fig.7). There's also no natural baseline provided, e.g. a dataset of the same size as the original (or until performance levels off) in order to understand how good this type of condensation can be in the limit.
>
> Thanks for the question. The baseline here is the work of (Zhao & Bilen, 2021) which we build on. For the setting with simplification loss weight 0, our method is identical to theirs, we also use their adapted code for it. So the accuracy with the highest bits per dimension can be considered a baseline. We added a note that without simplification loss, the dataset condensation is a reimplementation of  (Zhao & Bilen, 2021).

---

> > ### Author Response · Authors · 2021-11-16
> > **Thanks, comments below 2/2**
> >
> >
> > > Similarly, the results on training with simplified images (Fig.5) do not include baselines or other competing methods.
> >
> > There is no directly competing method that tries to simplify images in the input space using the bpds of a frozen generative model that we are aware of. The possibly closest related work is the cited work of Dubois et al. (2021), which however still differs substantially in motivation and methodology (trying to achieve maximum compression, main results using encodings of CLIP model, so not in input space, not focused on interpretability). Therefore, there is not really an appropriate baseline from prior work that we are aware of. Rather this work can be seen as establishing a baseline future work could compare to. Yet we very much like the suggestion to see what other baselines may be helpful, and we have now added one baseline/ablation to the supplementary section S8/Figure S4. We train a simplifier only on two losses, our bpd simplification loss and the mean squared error between the simplified and original images. Then, we retrain as before on the simplified images. Since no classifier is involved in the training of the simplifier here, this ablation checks whether the other parts of our during-training-loss actually improve retraining accuracies. As shown in supplementary, while the baseline also retains above-chance classification performance, it has a worse accuracy-bpd tradeoff showing our other losses lead to improved accuracies for the same bpd.
> >
> > **Update**: We have added another baseline where we blur the images using a gaussian blurring, again finding this leads to a worse accuracy-bpd tradeoff than SimpleBits, also see updated supp. section S8/Figure S4.
> >
> > > Fig.6: It would be interesting to see what the SimpleBits output is for the control image. For example, does the simplified image of the deer change considerably when the background color (outside the circle) is manually corrected? Are there cases where after correcting the suspected cause of error, the image is still misclassified? It would be good to show some unsuccessful/hard cases.
> >
> > These are very interesting questions! We have provided the SimpleBits output for the control images in Figure S15 in the supplement, where you can see reasonable features for the correct class. Also there are certainly hard-to-understand cases, we show an uncurated set of simplified images in supplement figures S16 and S17, feel free to let us know what you think of them.
> >
> > Sometimes correcting the suspected cause of error may not result in the expected prediction. This can happen for at least those reasons 1) the simplified image may contain a lot of the relevant information to relearn the prediction of the network, but might not show all of it, 2) the information that leads to the misprediction is not correctly identified by us from the simplified image, even though it is there 3) even if you remove the information that is crucial for the misclassification, there is no guarantee that the remaining information then must lead to the correct prediction, it can also lead to a prediction for yet another class. Therefore, that SimpleBits could be used to derive intuitions that could correct predictions can be seen as a good indication that the visualizations can help to understand the classifier better.
> >
> > > Sec.4: As a visual inspection tool, this is interesting, but the manual process of altering the images is unlikely to be viable for many applications like large datasets or streaming data). Do the authors anticipate a way of automating this process?
> >
> > The posthoc method could be extended by using different losses to produce the gradients, e.g., instead of kl-divergence to the original prediction use only the gradient of a specific class to see which information has been learned that increases the prediction for that class. In this way, one could show multiple SimpleBits images with different meanings automatically. Nevertheless, as any automatic interpretability method the output itself also needs to be interpreted. Here, interactive inspections can provide indications whether the intuitions gained from the method are correct or not. Note that while here we show post-hoc SimpleBits applied to misclassified images, it can just as well be applied to correctly classified images, see also the examples in supp. Figure S17. This can be especially interesting in the case of medical/scientific data where there is an increased interest in knowing the precise features predictive for a task and one may be willing to spend more time to investigate them. So we see the current best use for the method on problems where the decisions may be hard to explain, yet there is a great interest in understanding them.

---

> > > ### Comment · Reviewer_RfmX · 2021-11-22
> > > **-**
> > >
> > > I thank reviewers for the clarifications and extra results.
> > > I have updated the score.

---

> > > > ### Author Response · Authors · 2021-11-22
> > > > **Thanks; JPEG Baseline**
> > > >
> > > > Thank you very much for taking the time to read through our answers and updating your score.
> > > >
> > > > To address your comments about baselines from alternative image compression methods, we have added lossy JPEG compression with varying quality levels as another baseline for SimpleBits, again in section S8/figure S4. Please let us know if this is in the direction you would like to see to improve this manuscript or what other baselines you would consider interesting, also for future work on the manuscript.
> > > >
> > > > Thanks again for your time and effort.

---

### Official Review · Reviewer_ZQYq · 2021-10-31

**Correctness:** 3
**Technical Novelty And Significance:** 3
**Empirical Novelty And Significance:** 2
**Recommendation:** 5
**Confidence:** 4

**Main Review:**

Overall, the paper provides a diverse set of experiments showing that SimpleBits is able to reduce the image complexity in terms of bpd (bits per dimension) while retaining model performance at the same time. The proposed method is fully data-driven and does not have any strong assumptions.

**Strength**
- The proposed loss terms $\lambda_{sim} L_{sim} + L_{cls}$ are easy to implement
- The paper provides ways to apply the simplification network in various settings
- I like figure 4, where it is easy to see the progress of the simplification

**Weakness**
- Section 4 is hard to understand. What is the motivation for simulating forgetting? In Algo.1, what are $\mathbf{h}$ and $\mathbf{h}_{sim}$ in L4, 5? The gradients of the returned loss are taken w.r.t to which parameter, simplification network or $f$?
- In Section 5, it is unclear if the motivation here is to condense the dataset further or interpret neural nets.
- In the dataset condensation part, prior work (Zhao & Bilen, 2021) performs gradient matching in the inner loop and re-initializes the model parameters in the outer loop. Therefore, for each outer loop, a different neural network is used. It is unclear which model the proposed method is interpreting. In the rebuttal, the authors can explain the details of combining simplification and dataset condensation more.

**Other**
- For section 3 (simplification during training), can the proposed method identify spurious features learned by the neural nets?


**Summary Of The Paper:**

The paper proposes to minimize the image per-instance complexity to interpret the essential ingredients for neural network learning. The proposed method, SimpleBits, works in three different settings _i)_ Simplification during training _ii)_ Simplification after training _iii)_ Simplification with dataset condensation.

In _i)_, SimpleBits is able to differentiate task-relevant information on simulated composite datasets. In _ii)_, SimpleBits is able to explain the possible reason for misclassification. In _iii)_, SimpleBits reduces the image complexity in dataset condensation even further.

The major contributions can be summarized as follows.

- Simplifying images to interpret the neural models is not new, while this paper performs the novel simplification on image complexity.
- The paper proposes a way to simplify image complexity while retaining classifier accuracy.
- The paper shows that SimpleBits helps interpret the model learning in three different cases.

**Summary Of The Review:**

The paper proposes a novel direction of simplifying images (in terms of image complexity) to interpret neural networks. The proposed method is intuitive and easy to implement, having the potential to make broader impacts.
The paper applies the proposed, SimpleBits, in three different settings *i)* simplification during training *ii)* simplification after training *iii)* simplification with dataset condensation.

From the title, the paper's main goal should be understanding and interpreting neural networks, while I found the contents sway between interpreting models and simplifying images. Without further elaboration, better simplifying images does not naturally imply better model interpretation.

Overall, I think the method is novel, and the experimental results look convincing, but the explanation and motivation are unclear, especially sections 4 and 5.

---

> ### Author Response · Authors · 2021-11-16
> **Thanks, comments below**
>
> > * Section 4 is hard to understand. What is the motivation for simulating forgetting? In Algo.1, what $h$ are and $h_\mathrm{sim}$ in L4, 5? The gradients of the returned loss are taken w.r.t to which parameter, simplification network or $f$?
>
> The motivation for simulating forgetting is to find out which information in the image is crucial for the classifier network $f$ to retain or relearn its prediction, basically asking what information does the network need to see so that it can remember it’s prediction. In other words, we want to simulate a setting where the network has lost some of the information it has learned. And then we want to check which information it would relearn if it was trained to recover the original prediction on the original image. Using this, we want to ensure the network would relearn the same information from the simplified image. Information here refers to the values of the learned parameters. We scale the parameters down as this corresponds to losing information, needing less bits to encode the reduced parameters under a zero-mean prior on the weights (Hinton & van Camp, 1993). It is also a normal operation occurring in training as weight decay where it is meant to regularize the network. Also empirically it makes sense as it typically makes the predictions more uniformly distributed compared to the original network. Now, we have to define which information would be relearned on a specific image, so to what degree different original parameter values would be recovered from training to recover the original prediction. For this, we use the gradient of the KL divergence of the scaled networks and the original networks prediction, but only the part of the gradient that would lead the network closer to its original values.
>
> Here, $h$ is the prediction of the original network on the original image and $h_\mathrm{sim}$ on the simplified image. ${h}_\mathrm{scd}$ and $h_\mathrm{sim,scd}$ are the corresponding predictions for the scaled network. We added lines for $h$ and $h_\mathrm{sim}$  to Alg1.
>
> We compute the gradient with regard to the gates for the parameters of the *classifier* network $f$ as we want to see which information the classifier network would relearn (we only use the gradient that would lead closer to the original parameter values). This gradient for the original image defines the information that would be relearned. Now we try to ensure that the same information is relearned when training on the simplified image. We do so by penalizing the distance between gradients produced by the original and by the simplified images. There is no simplification network involved at all.
>
> Since several reviewers had trouble understanding this section, we have revised it to more clearly explain methodology and motivation. Thanks also for your questions which can help us in this revision process.
>
> > * In the dataset condensation part, prior work (Zhao & Bilen, 2021) performs gradient matching in the inner loop and re-initializes the model parameters in the outer loop. Therefore, for each outer loop, a different neural network is used. It is unclear which model the proposed method is interpreting. In the rebuttal, the authors can explain the details of combining simplification and dataset condensation more.
> > * In Section 5, it is unclear if the motivation here is to condense the dataset further or interpret neural nets.
>
> We follow their methodology, so as you said, we also reinitialize network parameters in the outer loop. So the method is interpreting what such a network *architecture* learns. At the same time, if you train any network on these condensed images, obviously the network can only learn what is in the images. We added a note that without simplification loss, the dataset condensation is a reimplementation of  (Zhao & Bilen, 2021). The motivation here is both, investigating the effect of further bit reduction and use the simplified dataset to interpret the information that can be learned from.
>
> > *  For section 3 (simplification during training), can the proposed method identify spurious features learned by the neural nets?
>
> In principle, this can be possible. We provide examples of potentially spurious features revealed by SimpleBits in a new supp. Section S10. As written there, these visualizations can be a starting point to investigate potentially spurious features, like whether these are also learned by normally trained classifiers or are an artefact of the SimpleBits training.
>
> Thanks four your questions, we hope to have clarified them, we would love to hear any further things that are still unclear.

---

> ### Author Response · Authors · 2021-11-23
> **Thanks again; further comments welcome**
>
> Thank you again for your review. Would be great to know if our responses have clarified your concerns and if the revision is moving the manuscript in the right direction from your point of view. Any feedback is appreciated also for future work on the manuscript.

---

### Official Review · Reviewer_agcx · 2021-11-02

**Correctness:** 3
**Technical Novelty And Significance:** 3
**Empirical Novelty And Significance:** 3
**Recommendation:** 6
**Confidence:** 3

**Details Of Ethics Concerns:**

No ethics concerns.

**Main Review:**

The proposed method has the potential to illuminate the important regions of an input image for classification while avoiding having to manually occlude parts of images to figure out the most relevant image parts. I especially like the idea of jointly optimizing the simplifier network and the classification network -- in this way, the simplifier network can explain the prediction made by the classification network, by showing the simplified version of the input image that retains the most salient features for classification, and we know that the classification network can perform reasonably well on simplified images. The integration of the simplification loss and the gradient matching loss for dataset condensation is also a great idea, because it allows us to not only condense a dataset but also illuminate the important aspects of each class in the dataset.

I have some questions/concerns regarding the use of SimpleBits for posthoc explainability analysis of an already trained model (Section 4). Is the goal to train a simplifier network to explain the prediction of an already trained image classifier (by showing a simplified version of the input image that contains the most relevant parts for classification)? I am not entirely sure from the reading of Section 4. Is Algorithm 1 (or Algorithm 2 in the supplement) trying to describe how one computes the loss function for training the simplifier network in this case? Do you then perform back-propagation to update the simplifier network? Why is scaling of parameters of the trained network needed? Also, what is the purpose/meaning of the gradients of the KL-divergence between the rescaled network's prediction and the original network's prediction? Section 4 needs a lot more explanations than what is currently written -- it needs to be rewritten for better clarity. Also, how does the proposed method compares with existing posthoc techniques (qualitatively or quantitatively)?

**Summary Of The Paper:**

In this paper, the authors proposed SimpleBits, which is a method to create simplified input images that retain the most relevant parts for classification while removing irrelevant details. The core of SimpleBits is the simplification loss, which is the negative log likelihood of seeing an image, according to an already trained generative model (as well as a related concept known as bits per dimension, which is simply the simplification loss divided by the dimension of a flattened image). The authors showed empirically that more complex images generally have higher bits per dimension/simplification loss.

The authors applied SimpleBits in three settings: (1) during training, (2) after training, and (3) dataset condensation/summarization. In the first setting, the authors trained both an image classifier and a simplifier network in tandem -- in particular, the simplifier network is trained by minimizing both the classification loss and the simplification loss. In the second setting, the authors tried to create simplified versions of input images, especially input images that are wrongly classified, to hypothesize (posthoc) about how an already trained network made predictions. In the third setting, the authors combined simplification loss with the gradient matching loss (Zhao and Bilen, 2021) to create a smaller dataset with simplified images, and showed that simplification does not have significant impact on accuracy.

**Summary Of The Review:**

The paper is for the most part well-written. Section 4 needs serious improvement in terms of clarity -- in particular, the authors should provide intuition and clearer explanations about how they performed posthoc explainability analysis using SimpleBits/a simplifier network (e.g., explanations about L_grad and L_pred in Algorithm 1 and why scaling of parameters of the trained network is needed). The authors should also consider comparing their methods of generating simplified images with existing posthoc techniques (e.g., saliency techniques). For a given input image, is the simplified image retaining regions similar to those identified as salient by an existing saliency method?

******************************
post-rebuttal:

Thanks for the clarification of Algorithm 1/2. After reading the response and the revised paper, I still think that the paper will benefit from a better explanation of how to synthesize a simplified image for a given input image for an already trained neural network. I understand that Algorithm 1/2 details the process of how to compute the loss function, but there is still a lack of explanation of how to optimize the loss function in the latent space. Essentially, I want to the following question be answered clearly: how do you perform the optimization in the latent space of the pretrained invertible generative model, using the loss function you computed in Algorithm 1/2? Right now, there is only an algorithm for how to compute the loss of the optimization problem, but I cannot find the optimization algorithm itself. Therefore, I am keeping my original score of 6.

---

> ### Author Response · Authors · 2021-11-16
> **Thanks, Comments below**
>
> > I have some questions/concerns regarding the use of SimpleBits for posthoc explainability analysis of an already trained model (Section 4). Is the goal to train a simplifier network to explain the prediction of an already trained image classifier (by showing a simplified version of the input image that contains the most relevant parts for classification)?
>
> Yes
>
> > I am not entirely sure from the reading of Section 4. Is Algorithm 1 (or Algorithm 2 in the supplement) trying to describe how one computes the loss function for training the simplifier network in this case?
>
> Yes, exactly it is the loss function for training. Added "function" to the Alg1 title to make it more clear. However, there is no simplifier network involved, see below.
>
> >  Do you then perform back-propagation to update the simplifier network? Why is scaling of parameters of the trained network needed? Also, what is the purpose/meaning of the gradients of the KL-divergence between the rescaled network's prediction and the original network's prediction? Section 4 needs a lot more explanations than what is currently written -- it needs to be rewritten for better clarity. Also, how does the proposed method compares with existing posthoc techniques (qualitatively or quantitatively)?
>
> Thanks for your precise questions. In Section 4, we do **not** use any simplifier network. Rather we directly synthesize the simplified images. One could do that in input pixel space, optimizing the individual pixels of the simplified images. However, we found it beneficial to perform the optimization in the latent space of our pretrained invertible generative model. So for a specific original input, we synthesize a  simplified input by optimizing a latent encoding. Now we need to optimize the latent encoding using losses defined in the input space. For that, we use the invertible network to invert the latent encoding into a simplified image in input space. In input space, we use the loss as described in Alg1 (in more detail, including the inversion step, in Alg 2). Again there is no simplifier network involved here at all. So yes, Algorithm1/2 describe the loss function but that loss is not used to optimize the simplifier network, rather it is used to directly optimize the latent encoding of the simplified image.
>
> Scaling of the parameters is needed as we look at the gradients of the loss of the KL-divergence of the scaled and the original prediction. Intuitively, we want to simulate a setting where the network has lost some of the information it has learned. And then we want to check which information it would relearn if it was trained to recover the original prediction on the original image. Using this, we want to ensure the network would relearn the same information from the simplified image. Information here refers to the values of the learned parameters. We scale the parameters down as this corresponds to losing information, needing less bits to encode the reduced parameters under a zero-mean prior on the weights (Hinton & van Camp, 1993). It is also a normal operation occurring in training as weight decay where it is meant to regularize the network. Also empirically it makes sense as it typically makes the predictions more uniformly distributed compared to the original network. Now, we have to define which information would be relearned on a specific image, so to what degree different original parameter values would be recovered from training to recover the original prediction. For this, we use the gradient of the KL divergence of the scaled networks and the original networks prediction, but only the part of the gradient that would lead the network closer to its original values. This gradient for the original image defines the information that would be relearned and now we try to ensure that the same information is relearned when training on the simplified image by penalizing the distance between gradients produced by the original and by the simplified images.
>
> Since several reviewers had trouble understanding this section, we have revised this section to more clearly explain methodology and motivation. We also added a note regarding $L_\mathrm{pred}$. Thanks again for your precise questions which helped in this revision process.
>
> > Also, how does the proposed method compares with existing posthoc techniques (qualitatively or quantitatively)?
>
> Thanks for this question. We have added a qualitative comparison to two other saliency-based methods in Figure 8 that also highlights more clearly what our method can uncover that saliency-based methods may struggle with.

---

> ### Author Response · Authors · 2021-11-23
> **Thanks again; further comments welcome**
>
> Thank you again for your review. Would be great to know if our responses have clarified your concerns and if the revision is moving the manuscript in the right direction from your point of view. Any feedback is appreciated also for future work on the manuscript.

---

### Official Review · Reviewer_eYVm · 2021-11-03

**Correctness:** 3
**Technical Novelty And Significance:** 3
**Empirical Novelty And Significance:** 3
**Recommendation:** 6
**Confidence:** 3

**Main Review:**

- Draft discusses an interesting problem. However, in the end-to-end learning framework it is likely to find task-irrelevant information in the input. While surely it is an interesting investigation to understand the amount of relevant information in the inputs, authors may better highlight practical utility of the simplified samples (or simplification).

- Since the simplified images have less irrelevant information how do they affect the time taken for training the models compared to the original samples?

- On the same lines, how can one leverage the trade-off between the model performance and sample simplicity? In other words, in the conventional training scenario the reader may like to know if there are any advantages/applications for sample simplification. For instance, do the simplified samples occupy lesser memory footprint than the original samples?

- Proposed per-instance simplification post model training can help to explore model behavior as a posthoc interpretation. However, it looks complicated to obtain as opposed to multiple gradient based posthoc explanations. It is not clear if they are related in some way.

- Shouldn't the notion of sample complexity be task specific? How is it justified to interpret the negative likelihood of the sample to be its complexity in the classification task?

- Authors use bpd as the image complexity measure which treats an image with more likelihood under a generative model trained on Tiny images dataset as less complex. From Figure 2 of the draft, a complete black image has close to 0 bpd, does that mean it has very strong likelihood? However, in general such images are not common in real datasets. Authors are suggested to clarify this.

- Also, how representative the computed likelihood can be, given that it is applied across different datasets (gray scale, color, etc. sometimes in different domains such as medical images) despite being learned on a specific dataset?

- In the experiment with the side-by-side MNIST dataset, SimpleBits output erases major portion of the relevant information also. I wonder what is the classification accuracy on the side-by-side data and the actual MNIST test data are.

**Summary Of The Paper:**

- Draft proposes a simplification method for visual recognition task where the information less relevant for the classification task is removed from the samples. In an iterative fashion the proposed approach (with no domain knowledge) synthesizes the simplified samples. Then the draft attempts to (i) measure the effect of the simplification via evaluating the models trained on the simplified samples, (ii) understand the models' behavior via posthoc interpretations, and (iii) simplify the synthetic data learned through dataset condensation.

**Summary Of The Review:**

- While the draft discusses a novel and interesting problem,  I feel it lacks in some of the aspects (listed in the Main Review). The work can be considered as a good early work in this direction. If the authors can justify the motivation for chosen complexity notion, and the effectiveness of the proposed posthoc interpretation along with other minor issues, this can become an excellent paper.

- Pre-rebuttal score: 5 (marginally below)
-----------------
post-rebuttal
----------------

- Rebuttal partially convinces me on some of the aspects (e.g. sample complexity) so I increased the score to 6.
- However, after reading other reviews and authors' rebuttal I feel that the manuscript needs some more improvements.
- For instance, primary aim of the manuscript is to investigate the effect of reducing information on classification, and I feel the findings are not very nontrivial or significant. In the end-to-end framework where the sample is presented to the learning algorithm as it is, one can easily see that he/she can find irrelevant information in the sample. And the framework deployed in the manuscript is very much inspired from the meta-learning approaches used for dataset distillation, condensation works. Also, the observations have not been leveraged (or harvested) in a significant way.
- Despite my earlier points, I feel that these investigations have potential to better understand the information that the classifier networks might be relying for learning. Hence, I score 6, but I will not fight for this manuscript in a discussion.

---

> ### Author Response · Authors · 2021-11-16
> **Clarifications and revisions 1/2**
>
> Thank you for your questions which were helpful to us in revising the manuscript for better understandability.
>
> > * Draft discusses an interesting problem. However, in the end-to-end learning framework it is likely to find task-irrelevant information in the input. While surely it is an interesting investigation to understand the amount of relevant information in the inputs, authors may better highlight practical utility of the simplified samples (or simplification).
> > * Since the simplified images have less irrelevant information how do they affect the time taken for training the models compared to the original samples?
> > * On the same lines, how can one leverage the trade-off between the model performance and sample simplicity? In other words, in the conventional training scenario the reader may like to know if there are any advantages/applications for sample simplification. For instance, do the simplified samples occupy lesser memory footprint than the original samples?
>
> Thank you for giving us the chance to further clarify the motivation of the current manuscript. It is correct that the simplified samples could occupy less memory and storage. The bpd values as shown on the x axis of Figure 5 reflect the minimum required storage given an optimal encoding scheme. But even for a generic encoder like PNG, the simplified images would require less storage space, leading to a tradeoff between storage and accuracy. We have added Figure S2 to the supplementary to showcase this tradeoff. Note however that these memory savings are not the core focus of the manuscript in its current form. Rather, the manuscript has two core aims: (1) investigate the effect of reducing information on the classification on different datasets and training settings as an interesting question in itself and (2) using these investigations to better understand the information that the classifier networks are using. Regarding training time, we have added a new supplementary section S7 with figure S3 with learning curves showing no substantial differences in training progress between more or less simplified images.
>
> > * Proposed per-instance simplification post model training can help to explore model behavior as a posthoc interpretation. However, it looks complicated to obtain as opposed to multiple gradient based posthoc explanations. It is not clear if they are related in some way.
>
> Indeed our method is computationally expensive and may be seen more complicated than some other post-hoc methods. The main advantage is that it can show more complex explanations than highlighting specific regions of the image or other more constrained explanations. If one would categorize post-hoc methods by the diversity of features they can visualize, from the most constrained ones being saliency-based methods that can only highlight image regions, to more flexible ones that learn a fixed set of features with a generative model (e.g. https://arxiv.org/abs/2104.13369), our post-hoc method could be considered to capture the most diversity: It can show any kind of feature for any kind of image, with the only constraint that the features have to be visualizable in a simplified image with few bits per dimension. To us, our post-hoc method is very promising to better understand some network decisions that other methods may be unable to provide any suitable explanation. A networks' decisions doesn’t have to rely only on specific regions in an input or some fixed set of features for every input. For example, consider a green global hue in the image may lead the network towards higher predictions for the frog class, which a saliency-based method might be unable to show. We have also added a qualitative comparison to two other saliency-based methods in Figure 8 to highlight these differences. The additional supp. Figures S15/16/17 also now showcase the diversity of features the post-hoc method can visualize.

---

> > ### Author Response · Authors · 2021-11-16
> > **Clarifications and revisions 2/2**
> >
> >
> > > * Shouldn't the notion of sample complexity be task specific? How is it justified to interpret the negative likelihood of the sample to be its complexity in the classification task?
> > > * Authors use bpd as the image complexity measure which treats an image with more likelihood under a generative model trained on Tiny images dataset as less complex. From Figure 2 of the draft, a complete black image has close to 0 bpd, does that mean it has very strong likelihood? However, in general such images are not common in real datasets. Authors are suggested to clarify this.
> > > * Also, how representative the computed likelihood can be, given that it is applied across different datasets (gray scale, color, etc. sometimes in different domains such as medical images) despite being learned on a specific dataset?
> >
> > Thank you for raising these questions which are quite central to the understanding of the manuscript. Multiple prior works had found that generative models trained on different image datasets learn characteristics common to all natural images (like local pixel correlations) and that these characteristics (e.g., degree of smoothness) dominate the log image densities (and, correspondingly, the bpds) computed by the generative models. See for example the works finding this in different ways when investigating anomaly detection: https://arxiv.org/abs/2102.08248 https://arxiv.org/abs/2006.10848 https://arxiv.org/abs/2006.08545  (and initial findings in https://arxiv.org/abs/1810.09136). In these works, this finding is often considered a problem for anomaly detection. A generative model trained on any image dataset may assign higher log-densities to simpler images like the black image you mentioned, even if such images would be considered out of distribution. Here, instead we try to treat this as a feature, using the generative model’s densities as a guide to synthesize simpler images. We have added part of these explanations to the manuscript and put an extension to Figure 2 in the supplement section S1, showing the bpds produced by other kinds of generative models trained only on CIFAR10 also order the images in the same way from visually simple to complex.
> > To answer the point about task-specificity, our bpd measure of complexity is not meant to be task-specific but rather a measure that correlates with general simplicity/complexity. We then use the other optimization objectives like our classification losses to find a tradeoff between general simplicity and task-specificity.
> >
> > > * In the experiment with the side-by-side MNIST dataset, SimpleBits output erases major portion of the relevant information also. I wonder what is the classification accuracy on the side-by-side data and the actual MNIST test data are.
> >
> > Good point, the accuracies are indeed a bit lower, originally our network achieves ~99% on the side-by-side dataset, it still retains ~96.5% on the simplified data.

---

> ### Author Response · Authors · 2021-11-23
> **Thanks again; further comments welcome**
>
> Thank you again for your review. Would be great to know if our responses have clarified your concerns and if the revision is moving the manuscript in the right direction from your point of view. Any feedback is appreciated also for future work on the manuscript.

---

### Author Response · Authors · 2021-11-16
**Thanks for your reviews; new revision to address your comments**

We thank all the reviewers for their reviews and detailed points which have helped us to revise the manuscript. We are happy the reviewers found that the first revision "discusses an interesting problem", has the potential "to become an excellent paper", describes "an intuitive method with a potential for broader impact", was "for the most part well-written" and provides an error-diagnosis "visualization tool that could be useful".

We made the following changes to the manuscript to address your comments; relevant changes are highlighted in dark green in the pdf:
* Revised abstract and parts of introduction to better explain the overall motivation
* Restructured the sections to a different order to improve the reading flow
* Added evidence that bpd can be a suitable measure of visual complexity. In addition to Glow, now we added PixelCNN and Diffusion model to the original Figure 2 (now Figure S1 in the supplementary section S1), showing bpds obtained from various generative models trained on various datasets also correlate with visual complexity.
* Expanded and revised the explanation of the motivation and methodology of simplification after training to improve understandability
* Clarified that the dataset condensation setting *without simplification* (loss weight 0) is just a reimplementation of prior work and therefore a fair baseline for SimpleBits to compare against
* Visualized some potential spurious features revealed by SimpleBits in Figure S8
* Added saliency method baselines -- Expected Gradients and Prediction Difference -- to Figure 8
* Added SimpleBits visualization of the control Image to the supplementary Figure S15, showing the simplified control images show reasonable features for the correct class. Also note we supply further uncurated post-hoc SimpleBits visualizations in supp. Figures S16 and S17.
* Added baselines for per-instance simplification during training. We 1) apply a Gaussian blurring filter to manually simplify images with increasing intensities, 2) JPEG Compression with varying quality levels and 3) learn simplification with two simple losses. Then we retrain classifiers with those simplified images. In all cases the accuracy-bpd tradeoff is worse than SimpleBits. Please see updated supp. section S8/Figure S4.

We have addressed further comments in the individual answers to your reviews and added several supplementary sections to address your questions. We hope these revisions have addressed your concerns and would love to hear further feedback.

---

### Decision · Program_Chairs · 2022-01-20

**Decision:**

Reject

**Comment:**

This paper proposes SimpleBits, for simplifying input images to remove irrelevant details but keep relevant details for classification. This idea can be applied during/after training. Authors have significantly revised the draft to address reviewer concerns, to improve the readability and clarify concerns on complexity analysis, for which reviewers have raised scores post-rebuttal. However, even with score changes, there are commonly expressed concerns, that manuscript still needs some more improvements to be ready for publication in their post-rebuttal comments: findings are not very nontrivial or significant (reviewer eYVm), still incomplete (RfmX) or optimization algorithm is yet to be found (reviewer agcx) .